Corrected: Author correction

# Intracellular localization of nanoparticle dimers by chirality reversal

Maozhong Sun[1,2], Liguang Xu[1,2], Joong Hwan Bahng[3,4], Hua Kuang[1,2], Silas Alben[5], Nicholas A. Kotov [3,4,6,7,8] & Chuanlai Xu [1,2]

The intra- and extracellular positioning of plasmonic nanoparticles (NPs) can dramatically alter their curative/diagnostic abilities and medical outcomes. However, the inability of common spectroscopic identifiers to register the events of transmembrane transport denies their intracellular vs. extracellular localization even for cell cultures. Here we show that the chiroptical activity of DNA-bridged NP dimers allows one to follow the process of internalization of the particles by the mammalian cells and to distinguish their extra- vs intracellular localizations by real-time spectroscopy in ensemble. Circular dichroism peaks in the visible range change from negative to positive during transmembrane transport. The chirality reversal is associated with a spontaneous twisting motion around the DNA bridge caused by the large change in electrostatic repulsion between NPs when the dimers move from interstitial fluid to cytosol. This finding opens the door for spectroscopic targeting of plasmonic nanodrugs and quantitative assessment of nanoscale interactions. The efficacy of dichroic targeting of chiral nanostructures for biomedical applications is exemplified here as photodynamic therapy of malignancies. The efficacy of cervical cancer cell elimination was drastically increased when circular polarization of incident photons matched to the preferential absorption of dimers localized inside the cancer cells, which is associated with the increased generation of reactive oxygen species and their preferential intracellular localization.

[1] State Key Laboratory of Food Science and Technology, Jiangnan University, Wuxi , 214122, China. [2] International Joint Research Laboratory for Biointerface and Biodetection, Jiangnan University, Wuxi , 214122, China. [3] Chemical Engineering Department, University of Michigan, Ann Arbor, MI 48109, USA. [4] Department of Biomedical Engineering, University of Michigan, Ann Arbor, MI 48109, USA. [5] Department of Mathematics, University of Michigan, Ann Arbor, MI 48109, USA. [6] Department of Material Sciences and Engineering, University of Michigan, Ann Arbor, MI 48109, USA. [7] Michigan Center for Integrative Research in Critical Care, Ann Arbor, MI 48109, USA. [8] Biointerfaces Institute, University of Michigan, Ann Arbor, MI 48109, USA. Maozhong Sun and Liguang Xu contributed equally to this work. Correspondence and requests for materials should be addressed to H.K. (email: kuangh@jiangnan.edu.cn) or to N.A.K. (email: kotov@umich.edu)

Plasmonic nanoparticles (NPs) and nanorods (NRs) are the basis of several propitious techniques for disease diagnostics and treatment, as exemplified by photodynamic/photo-thermal therapy[1–4], gene/protein delivery[5, 6], hyperspectral imaging[7–11], and small interfering RNA (siRNA)/microRNA (miRNA) medicine[12, 13]. Intracellular localization of NPs is paramount for eradication of malignancies and other cellular abnormalities based on plasmonic effects or drug delivery: the treatment efficacy can exhibit order of magnitude differences[12, 14]. Although spectroscopic localization of single particles with sub-cellular accuracies is possible[10, 15], there are no ensemble methods for the conclusive determination of whether NPs added to cell culture or tissue are inside or outside the cells. For instance, the dependence of plasmonic extinction of individual gold NPs and NRs on dielectric constant could potentially allow differentiation of extracellular and intracellular media by their dielectric constant[16]; however, the shift of the plasmonic bands is small. In addition, the intensity and the width of plasmonic bands is strongly affected by cellular scattering, protein adsorption, and tissue heterogeneity[17–21]. Surface-enhanced Raman scattering (SERS) is a versatile tool for label-free biochemical analysis, but high-power lasers perturb and rupture live cells[22]. Both quenching and enhancement of luminescence can result from exciton-plasmon resonance between dyes and NPs[23–25], which makes emission of NP conjugates with cell-reporting dyes difficult to interpret. Cellular auto-fluorescence in the visible spectrum and photo-bleaching of fluorescent dyes further complicate the task. Hyperspectral imaging[7, 10, 15, 17, 20, 26] based on point-spread functions deconvolution can identify the geometrical position of the single particle in respect to the intracellular compartments[7] or cell membrane[27]. They afford nanometer-scale accuracy of NP localization in single specific cells but not in ensemble or complex inhomogeneous media.

Finding a distinct spectroscopic signature of transmembrane transports for plasmonic and other NPs will provide a new tool for both medical research and perhaps also clinical practice. Recent studies have revealed strong chiroptical activity for NPs and their assemblies[28–31]. The signal-to-noise ratio (SNR) specific to chiroptical activity measurements in the region of gold plasmons enabled a marked reduction of the limit of detection in biosensing (3.7 aM of dsDNA strands or ca. 1100 assembled NRs in a sample)[32]. Since the geometry of NP assemblies with DNA connections can be dynamic[33–35], we hypothesized that the intense chiroptical activity of NP/NR dimers, and the flexibility of the DNA bridges between them, may lead to distinctive readout for events when NP traverse cellular membranes due to protein adsorption and differences in the chirality of the molecular components in the intracellular vs. interstitial spaces (Fig. 1a). It was found that chiroptical measurements can indeed provide a powerful real-time spectroscopic technique for monitoring transmembrane transport, although the mechanism of this effect revealed differences with the original hypothesis.

## Results

**Assembly of chiroplasmonic dimers.** In this work, we used dimers[32] of slightly oblong Au NPs with average lengths and widths of the metal core of $22 \pm 3$ and $17 \pm 1$ nm, respectively (Supplementary Figs. 1–3). These particles were coated by thiol-modified polyethylene glycol-5000 (SH-PEG-5000) in the amount of $860 \pm 60$ molecules per NP. The thickness of PEG coating was ca. 5 nm and is sufficient to reduce van der Waals attraction of proteins to metal core of the NP and, therefore, reduces the influence of the biomolecules around the particles on their optical properties[36]. Coating NPs with PEG-5000 camouflages them

from detection by reticuloendothelial system[37, 38] being assisted by formation of the soft biomolecular corona[39–42].

NP coating also included cell penetrating peptides (TAT) in the amount of $190 \pm 16$ peptide ligands per NP in order to facilitate the virus-like transport through the cellular membrane[43, 44]. Addition of TAT peptides enables NPs and their constructs to avoid endocytosis, endosomal segregation, and direct penetration into cytosol[43, 45].

We also used Au NRs with a length of $59 \pm 4$ nm and a diameter of $24 \pm 2$ nm as benchmarks. Both types of dimers–derived from NPs and NRs—are bridged by double-stranded DNA (dsDNA, 5′-CAATAGCCCTTGGAT-3′ and 5′-ATCCAAGGGCTATTG-3′) and have chiral scissor-like nanoscale geometry with dihedral angles of 5–15 degrees[46] between the long axes of the plasmonic particles, leading to strong peaks in the circular dichroism (CD) spectra. The 5′-AAAAAAAAAA-SH-3′ end sections anchor the DNA bridges to the metal surface both through Au-S covalent bond and intermolecular attraction between adenines and Au NP surface. The number of DNA strands between the NP does not exceed $1.6 \pm 0.2$[47]. The CD spectra of the NP dimers in 0.01 M phosphate buffer (pH 7.4) display distinct chiroptical peaks positioned at 494 nm (positive) and 522 nm (negative) (Fig. 1b).

**Chirality reversal of chiroplasmonic dimers.** Cervical cancer HeLa cells do not exhibit intrinsic chiroptical activity in the 400–900 nm spectral region (Fig. 1b, Supplementary Fig. 4). Their incubation with NP dimers results in the gradual reversal of the polarity of CD peaks at 500 and 530 nm but retention of their spectral positions (Fig. 1c–e, Supplementary Fig. 7). The most significant change in chiroptical activity takes place over the first 2 h and continues for the next 48 h (Fig. 1e). Removal of extra-cellular NP dimers shows that only the NP dimers inside the cells are responsible for the negative CD peak at 508 nm and positive CD peak at 537 nm (Fig. 1d, e). The shift of the chiroptical extrema is ascribed to the differences in the environment around the dimers inside and outside of the cells. The kinetics of the chiroptical effects is in line with previous studies of virus-like behavior of TAT-modified NPs[43, 45]. The direct transmembrane transport of NP dimers was detected by biological transmission electron microscopy (bio-TEM), which showed the particles to be dispersed in cytosol and not clustered in vacuoles (Fig. 2a, b) until 24–48 h had elapsed (Supplementary Figs. 9 and 10).

Importantly, the ultraviolet–visible (UV–vis) absorption peaks of NPs under the same conditions displayed little change (Supplementary Fig. 5). Despite the fact that NP agglomeration inside the cells is conducive to the shift of UV–vis plasmon peaks due to alterations in dielectric conditions around the clusters[9, 48], broadening of the peaks makes it difficult to register them accurately (Supplementary Fig. 6). SERS spectra, however, have much sharper bands and their intensity does change in the course of the experiment (Supplementary Fig. 8); the intensity of the SERS band at $\sim 513$ cm$^{-1}$ gradually increases and reaches a maximum after 24 h.

Cryo-TEM tomography and statistical analysis of tomographic images (Fig. 2c, d) establish that the NP dimers reconfigure from the right-handed conformation to the left-handed one upon transport into the cells: the average dihedral angles inside and outside the cells are $-12.5 \pm 2$ deg and $+9.6 \pm 1$ deg, respectively. Simultaneously, a large change of the distance between adjacent NPs can be seen in cryo-TEM images, with the gap expanding from $5.4 \pm 0.2$ nm in the buffer to $8.8 \pm 0.1$ nm inside the cells (Supplementary Figs. 13 and 14).

Knowing the dimer geometry, the chiroptical spectra of NP dimers can be calculated for intra- and extracellular localization

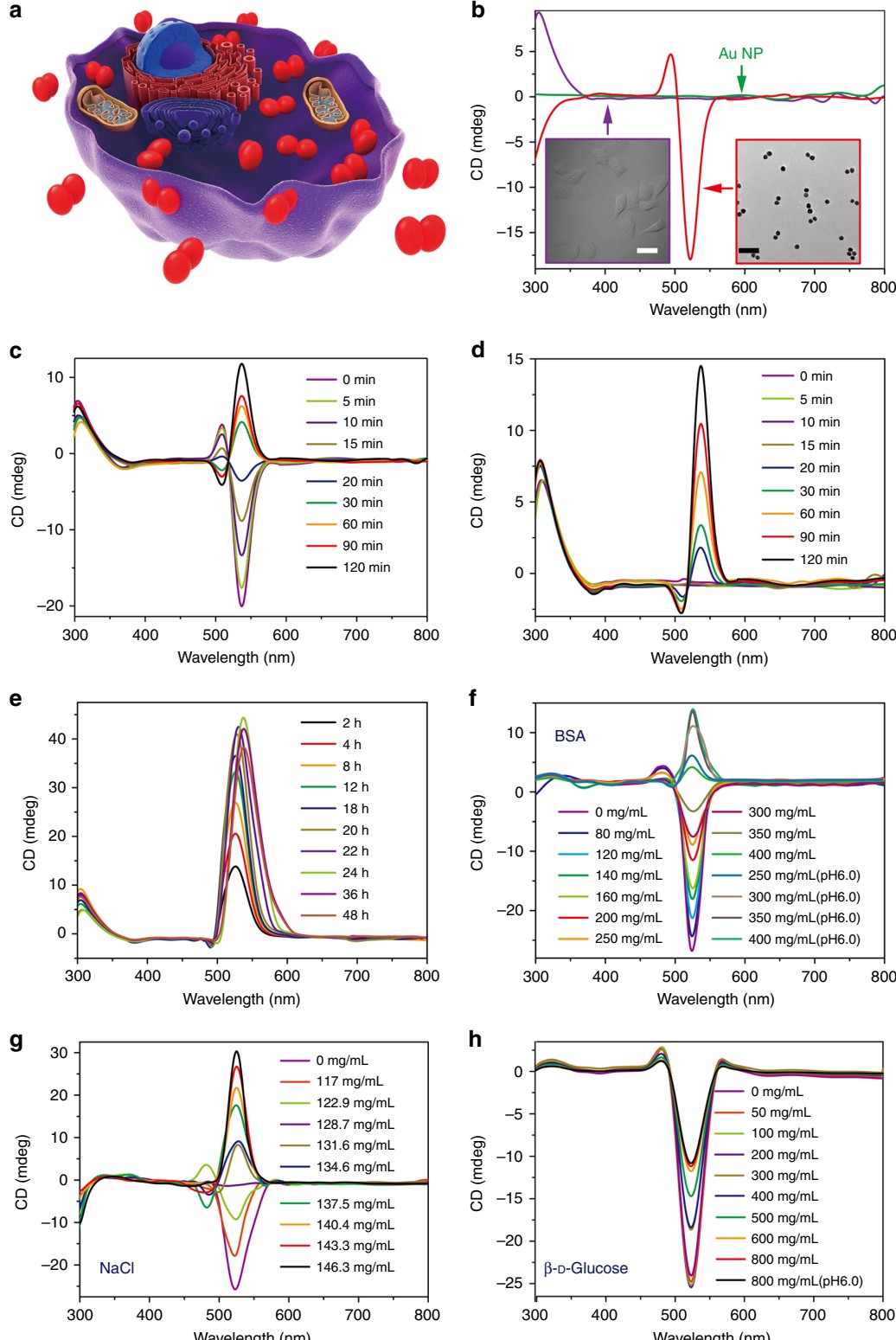

**Fig. 1** Temporal progressions of chiroptical activity of NP dimers for transmembrane transport in HeLa cells and other conditions. **a** Schematics of NP dimers in a model cell; **b** CD spectra of individual NPs, HeLa cells and NP dimers in the PBS buffer. Scale bar: 100 nm. **c**, **d** CD spectra of NP dimers incubated with HeLa cells over a period of 2 h; The spectroscopic measurements for each time point were performed before and after the removal of extracellular dimers. **e** CD spectra of NP dimers incubated with HeLa cells over a period of 48 h; The spectroscopic measurements for each time point were performed after the removal of extracellular dimers as in **d**. **f**–**h** CD spectra of NP dimers in BSA, NaCl and β-D-glucose solutions with various concentrations as given in the legends

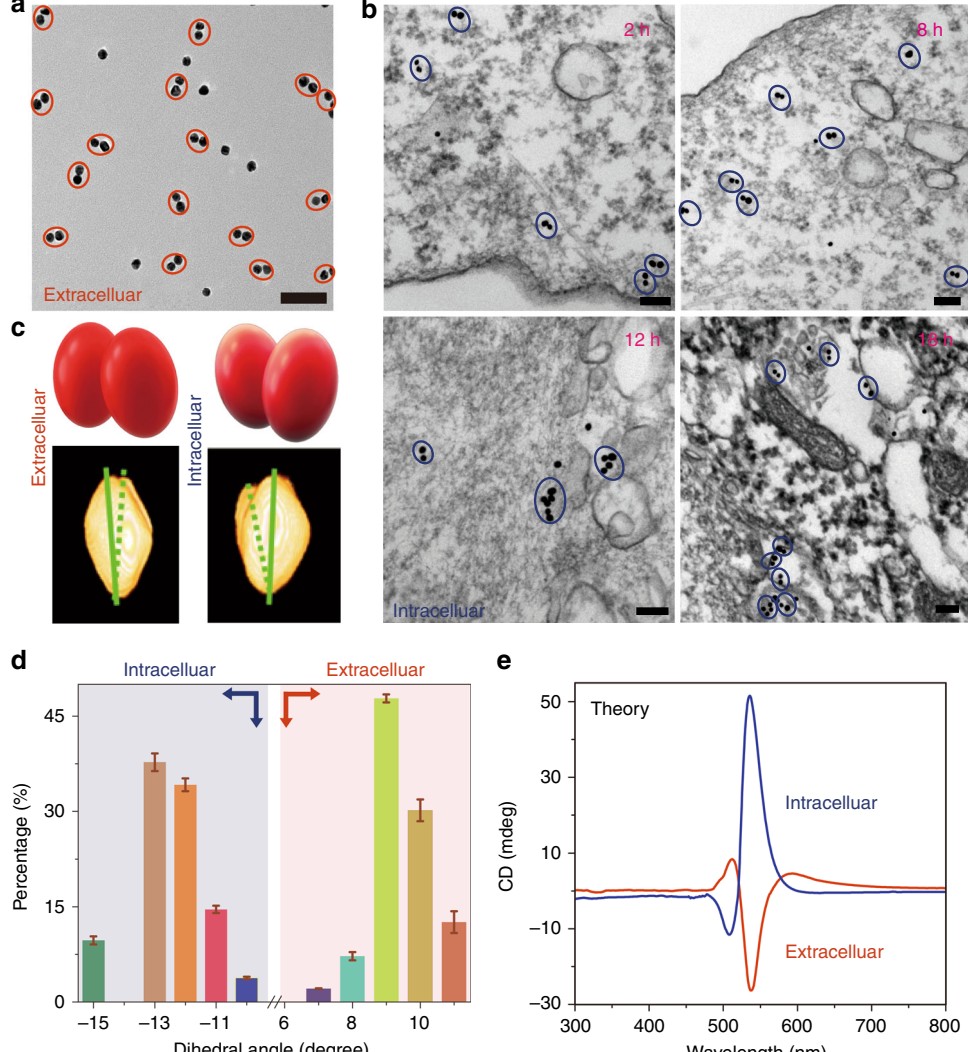

**Fig. 2** Chiral geometry of NP dimers. **a** TEM image of NP dimers in cell culture media; Scale bar: 100 nm. **b** Bio-TEM images of NP dimers in the HeLa cells; Scale bars: 100 nm. **c** TEM tomography images (bottom) of NP dimers both outside and inside cells with schematics of dimers' geometry (top). **d** Statistical analysis of the dihedral angles θ for NP dimers inside and outside the cell as determined from cryo-TEM tomography images. The error bars correspond to the standard error of the mean (n = 3). The sign of the dihedral angle in these nanoscale structures was chosen in accord with the IUPAC convention. **e** Simulated CD spectra of NP dimers intra- and extracellular localization of NP dimers based on geometries from **d**

of the dimers. The simulated CD spectra match the experimental data for the dimers both inside and outside the cells nearly perfectly (Fig. 2e). Besides verifying the conformational switching of the NP pairs, computational tools provide a path to quantitative analysis of chiroptical data and the percentage of chiral tags internalized by the cells. These calculations also highlight the stark difference in sensitivity of CD vs. UV–vis spectra to the change of the conformation and environment around the NP dimers (Supplementary Figs. 18 and 19).

To further substantiate the significance of the dimer twisting for understanding the chirality reversal mechanism, we made dimers from NPs and NRs that lack ability to rotate around the DNA bridge as a negative control for the previous experimental series. Dimers from NPs and NRs were coated with an amphiphilic diblock copolymer polystyrene-block-poly(acrylic acid) (PS-PAA, Fig. 3a, c, Supplementary Figs. 20, 21 and 24–27), which adsorbs to NP assemblies[49] and impedes the ability of the dimer to reconfigure. When the PS-PAA-encapsulated NP dimers are incubated with HeLa cells, they undergo transmembrane transport (Fig. 3b, Supplementary Figs. 29 and 30) but the

overall intensity and polarity of the CD peaks exhibit no change for all incubation times both for NP and NR dimers (Supplementary Figs. 22, 23 and 28).

**Role of plasmonic hotspots.** Based on theoretical work of Zhang et al.[36] and experimental work of Lieberman et al.[50], chiroptical activity of DNA-bridged NP assemblies have been discussed in the recent literature in context of plasmonic hotspots. Their role in chiroptical activity and chirality reversal was also considered in this study. We found that the hotspot-enhanced circular dichroism plays a minor role in the chiroptical activity of our dimers, because the gap between the NPs is larger than what is needed to have electrical field of sufficiently high intensity. A computer simulation with the full space integration (4π ster) for two NP dimers bridged by a DNA-like helix was carried out (Supplementary Fig. 17). It was found that the chiroptical activity attributable to hotspots is ~ 10x smaller than that originating from the twisted geometry of the dimer.

The evidence of chiroplasmonic activity in our dimers has little relation to the hotspots can also be found in the experiment. It is

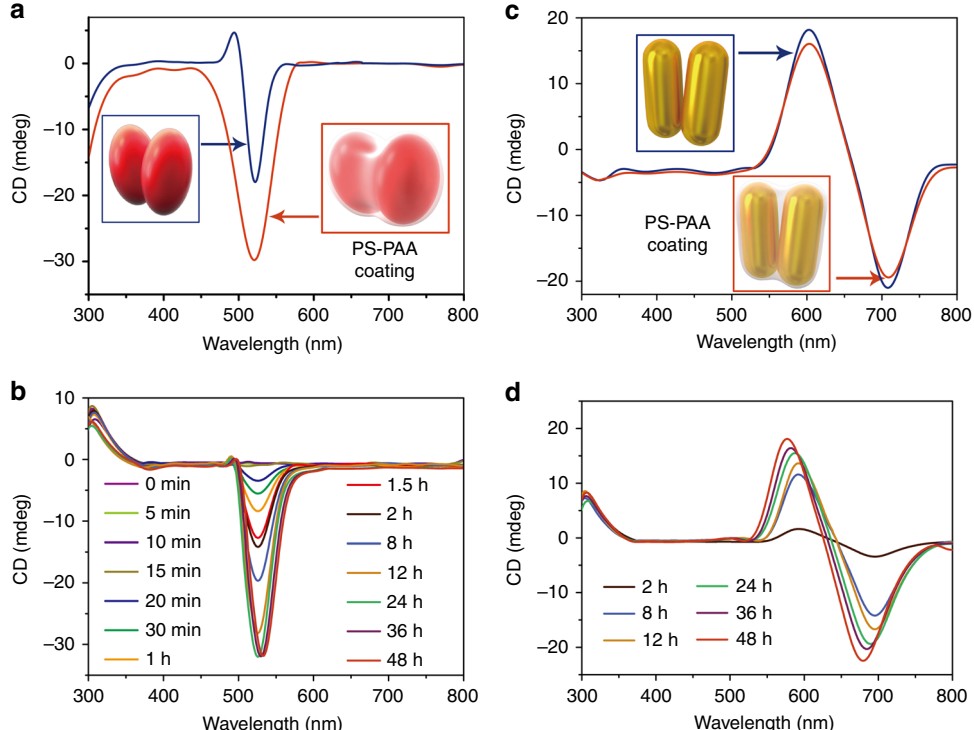

**Fig. 3** Transmembrane transport and chiroptical activity of NP (**a**, **b**) and NR (**c**, **d**) dimers with PS-PAA shells with stationary conformation. **a** CD spectra for NP dimers in buffer before and after encapsulation by PS-PAA and (**b**) their temporal progression upon incubation with HeLa cervical cancer cells indicating the gradual cellular incorporation but no change in the sign of the chiroptical activity. The spectroscopic measurements at each time point were performed after the removal of extracellular excess of NP dimers. Statistical analysis of the dihedral angles for NP dimers before and after encapsulation by PS-PAA is given in Supplementary Fig. 23. **c** CD spectra of NR in buffer before and after encapsulation by PS-PAA, and (**d**) their temporal progression upon incubation with HeLa cells; the spectroscopic measurements at each time point were performed after the removal of extracellular excess of NP dimers.

well known that the hotspots are responsible for enhanced Raman scattering[51, 52]. However, the temporal progression of Raman spectra (Supplementary Fig. 8) does not correlate with transmembrane transport of the dimer and the temporal progression of CD spectra (Fig. 1c, d). The dynamics of SERS and chiroplasmonic peaks have characteristic times that are different from each other by more than one order of the magnitude (~50 times), which indicates that they have different origin. Additional discussion of the hot spot mechanism as applied to the NP assemblies used in this work is given in Supplementary Information.

**Role of dielectric environment.** The complexity of the interactions between the NP dimers inside and outside of the cell is staggering. Multiple proteins and other biomolecules could be adsorbed on the dimers. Even in a comparatively simple case of dimers in the buffer the accurate quantification of the forces between the NP assembled in the nanoscale dimer encounters multiple hurdles[53]. Currently neither extended Derjaguin-Landau-Verwey-Overbeek (DLVO) theory nor simulation techniques, such as Molecular Dynamics (MD), can properly quantify these interactions. Notwithstanding these fundamental challenges associated with both nanoscale state of matter and composition of biological media, it is important to identify physical concepts behind the experimental observations of chirality reversal. Even limited order-of-magnitude quantification helps in understanding the mechanism of chiroptical phenomena here and therefore, further engineer such systems.

Let us consider NPs interacting through a heterogeneous medium having an effective dielectric constant. The difference in dielectric conditions inside and outside of the cells is massive. The ionic strengths in extracellular space and cytosol of HeLa cells are 0.026 and 0.18 mol/l, respectively[54]. Protein concentration reaches 400 mg/ml inside HeLa cells[55], while it is one order of magnitude smaller, i.e., 40 mg/ml, outside the cells and in the buffer.

The fact that the observed chirality switching may originate from the dielectric environments can be discerned from the effects of NaCl, D-glucose, and bovine serum albumin (BSA) on CD spectra of the dimers (Fig. 1) (Supplementary Information).

Note also that NaCl concentration and dielectric constant of the media have a larger effect on the conformation of the dimer than the concentration of D-glucose and BSA (Fig. 1f–h and Supplementary Figs. 11, 12, 15 and 16) indicating that electrostatic interactions must play a significant role in the dynamics of the chiroptical properties.

The equilibrium conformation of the NP dimer must be determined then by the balance of electrostatic repulsive and other attractive forces originating both in DNA bridges and NPs mediated through the surrounding environment. A minimalistic description of such system includes the energy contributions from electrostatic repulsion of NPs ($W_{el}$), the elastic deformation of spring-like DNA bridges ($W_{spr}$), van der Waals attraction ($W_{vdW}$) and hydrogen bonds ($W_h$). Partially relaxing non-additivity problem of DLVO by including an additional parameter of dihedral angle, $\theta$, to complement the separation distance $h$ typically considered in DLVO, $W_{total}$ will be represented as a function of

$$W_{total} = W_{el}(\theta, h) + W_{spr}(\theta, h) + W_{vdW}(\theta, h) + W_h(\theta, h), \quad (1)$$

For our dimers connected by the stretched DNA bridge, electrostatic repulsion is countered by the restoring force from elastic deformation of the DNA "springs", hydrogen bonds, and vdW forces. $W_{el}$ of two elongated NPs (or NRs) with long axes forming a dihedral angle, $\theta$, and separated by the distance $z$ can be estimated by the line charge approximation (Supplementary Eq. 9, 10 and Supplementary Fig. 31). Unfortunately there is no set

of fully defendable dielectric parameters that include dielectric constant, surface charge, ionic strength in the nanoscale gap, etc. Furthermore, for intracellular environment even the dielectric constant, $\varepsilon_0$, varies from 15 to 60 according to different studies[56, 57], which adds challenges to the electrostatic energy calculations. Given these uncertainties, we estimated the likely range of electrostatic energy $W_{el}$ using line charge approximation

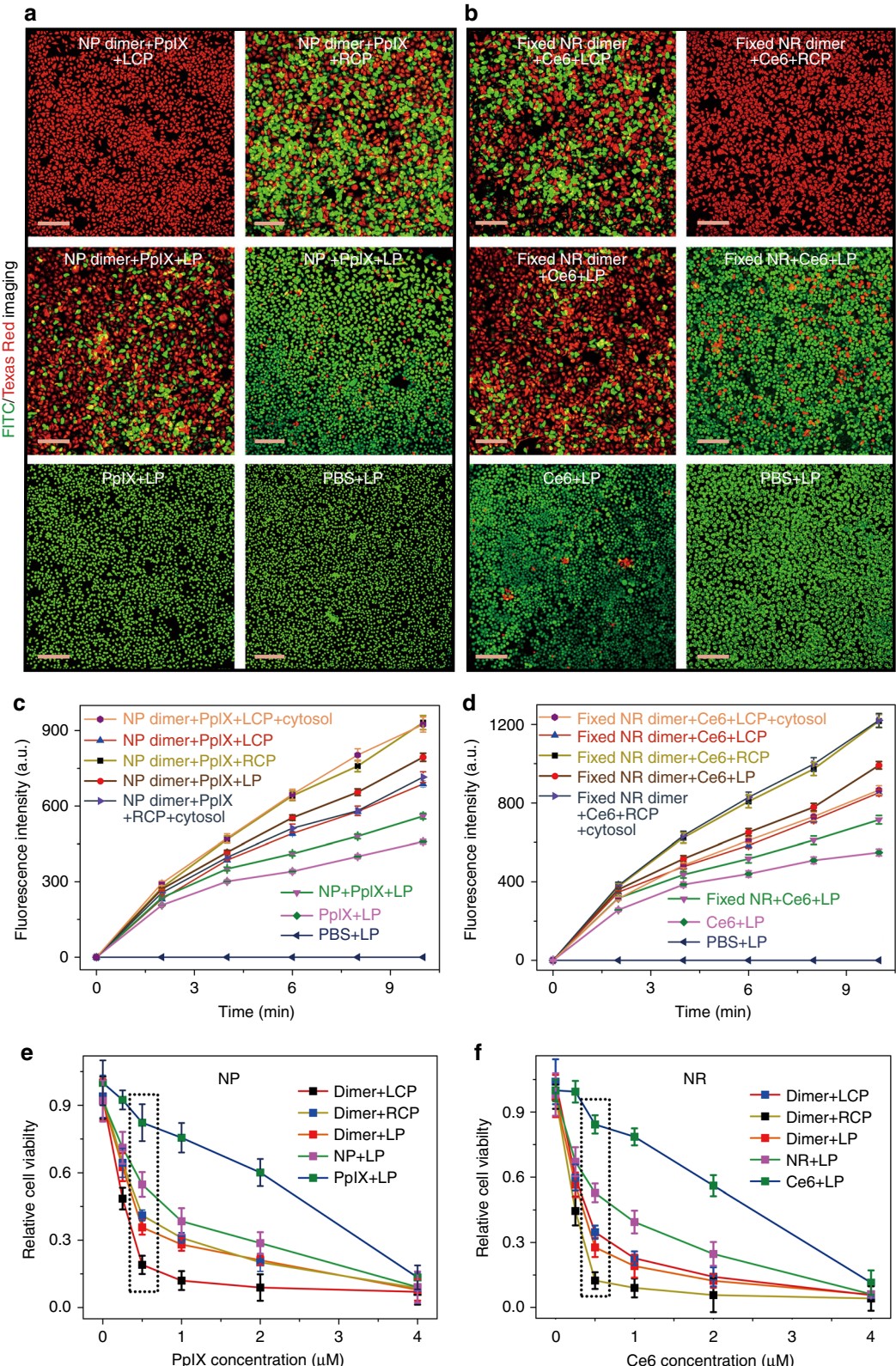

described in Supplementary Fig. 31, which came out to be from $5\times10^{-19}$ J to $5\times10^{-20}$ J inclusive of intracellular and extracellular environments.

The spring-like properties of dsDNA are described by a twist-stretch model[58] with the elastic energy equal to:

$$W_{spr} = \left(\frac{c\theta^2}{2L} + \frac{g\theta}{L}\chi + \frac{S}{2L}\chi^2\right), \qquad (2)$$

where $L$ is the unstrained length of dsDNA, $x$ is the distance that the dsDNA is stretched beyond the unstrained length, $c$ is the twist rigidity, $g$ is the twist-stretch constant[58] and $S$ is the modulus for stretching deformation[58]. Taking into consideration various values of these constants obtained from experimental studies[14, 59, 60] and MD simulations[61], one can estimate the range of the $W_{spr}$ to be from $2\times10^{-19}$ J to $4\times10^{-19}$ J inclusive of both intracellular and extracellular environments.

Using Hamaker approximation, one can estimate $W_{vdW}$ to be between $7\times10^{-20}$ J and $1\times10^{-19}$ J. Although similar order of magnitude estimates can be also obtained for $W_h$, a conceptual conclusion can be made based on these estimates. One can see that $W_{spr}$ represent a large if not dominant component of the Eq. 1 which will result in strong dependence of the $W_{total}$ on $\theta$ according to Eq. 2. So, the preference to acquire a scissor-like conformation is likely to originate from the elastic force exerted by DNA binds the particles together but also results in a twisting motion and strong $\theta$-dependence due to the helical conformation of the DNA bridge. Even if elastic force is not the dominant one, the angle dependence of the electrostatic force for elongated NPs (Supplementary Fig. 31) will also be present. Therefore, a minimum of $W_{total}$ and the preferred geometry of the dimer will be found for positive and negative $\theta$ depending on the chemical and dielectric properties of the dimer's environment. The dimers with positive and negative $\theta$ correspond to the negative and positive CD peaks at 530 nm in Fig. 1, respectively.

**Chiroplasmonic photodynamic therapy**. The strong chiroptical activity of NP dimers makes conformation reversal relevant to photodynamic and photothermal therapies[62] and also potentially to other medical techniques[63]. Chirality reversal upon trans-membrane transport should enable dichroic targeting of photodynamic therapy or, by other words, the dependence of apoptosis induction efficacy on the circular polarization of light. To test this hypothesis, we illuminated HeLa cells incubated with (Figs. 4 and 5) and without (Supplementary Fig. 33) NP dimers carrying protoporphyrin IX (PpIX, Supplementary Fig. 32) photosensitizers by 532 nm light under different polarization conditions. These photons match the absorption band of PpIX and the strongest CD band of the NP dimers (Supplementary Fig. 34)[64]. The chiroplasmonic band is positive for intracellular localization of the NP dimers (Fig. 1e). Hence, left-handed circularly polarized light (LCP) gives a three-fold higher apoptotic index in

cancer cells compared with right-handed circularly polarized light (RCP) or linearly polarized light (LP), Fig. 4a.

To confirm the effect of circular polarization in the eradication of cancer cells, a similar experiment was also carried out with stationary PS-PAA-encapsulated NR dimers carrying a Chlorin e6 (Ce6) photodynamic therapy agent (Supplementary Figs. 35–37). 660 nm photons were used in this case to match the absorption bands of Ce6 and NRs[65]. In the full agreement with the spectroscopic observations in Fig. 3d showing that the CD peak for this wavelength is negative, RCP light gives a higher percentage of apoptotic cells than LCP due to its stronger absorption by NR dimers (Fig. 4b). Comparison of free radical amounts produced by photosensitizers under similar conditions confirms dichroic targeting of cancer cells incorporating of chiral dimers with specific handedness (Fig. 4c, d).

Notably the difference between LCP and RCP illumination in apoptosis induction both for NP dimers and NR dimers was much greater than one might expect based on the intracellular concentration of photosensitizers, g-factor, and the difference of light absorption by the twisted rod pairs. This is quite surprising from the optical standpoint and a detailed study needs to be carried out to fully clarify the origin of such strong photodynamic effect. Based on the current data it is evident that rate and localization of reactive oxygen species (ROS) generation are the key in understanding the biological effect of RCP and LCP photons. Particularly informative is the fact that the experimental rate of ROS generations for NP and NR dimers in Fig. 4c, d changes concomitantly with the killing rate in. Based on the cast body of the previous studies on the plasmon-exciton dipolar coupling[24, 25] one can also expect that the dimers not only carry PpIX and Ce6 inside the cell but they enhance ROS generation potentially via energy transfer from NPs/NRs to photosensitizers.

The large difference in photodynamic activity of RCP and LCP photons also originates from the fact that the concentration of ROS in the process of photodynamic therapy must be over the critical threshold to induce cell death[66]. Consequently, apoptosis induction depends on the transient concentration of ROS and therefore, on light absorption, in a highly non-linear fashion. The large difference between RCP and LCP photons is specific to the concentration of the photodynamic therapy agents and the chiral dimers. If one chooses the concentration of 4 μM for PpIX/Ce6, almost all of the cell will be killed regardless of any light used. Similarly, if the low concentration (0.25 μM) of PpIX or Ce6 is used, the difference of death ratio between RCP, LCP, and other polarization will be small again (Fig. 4e, f). In order to obtain the optimum therapeutic effect and large difference between polarization states of incident light, the appropriate concentration of PpIX or Ce6 needs to be used that is (e.g., 0.5 μM in Fig. 4e, f). Then, the difference in absorption of circular polarization of photons is amplified by the threshold phenomena.

Short life time of ROS and localization of the dimers are also essential in understanding the LCP/RCP difference. The life time

**Fig. 4** Dichroic targeting using dimer chirality and photodynamic therapy. **a, b** Live (green, FITC)/dead (red, Texas Red) assays with confocal microscopy for adherent HeLa cells after 30 min illumination under different polarization conditions for NP dimers (with cell penetrating TAT peptides on the NP surface) with 532 nm photons (**a**) and NR dimers (with cell penetrating TAT peptides on the NR surface) with 660 nm photons (**b**) with variable and stationary conformations, respectively. **c, d** Ex vivo singlet oxygen generation as a function of light exposure in model dispersions of NP dimers **c** or NR dimers **d** conjugated to PpIX and Ce6 photosensitizers with variable (as in Figs 1c–e) and stationary (labeled as 'fixed') conformations (as in Fig. 3), respectively. 5 mW/cm² light at 532 nm for 30 min was used for NP dimers; 5 mW/cm² light at 660 nm for 30 min was used with stationary NR dimers. ~94 molecules of PpIX were conjugated onto each NP and ~116 molecules of Ce6 were conjugated onto each NR. Intracellular conditions for **c, d** were experimentally reproduced by model cytosol medium as described in the Supplementary Methods; error bars are given for standard deviation of 95%. **e, f** Viability of cells for different illumination conditions in the presence of various cellular loadings of PpIX (**e**) and Ce6 (**f**). The concentrations of photosensitizers were calculated in accordance with the average number of PpIX and Ce6 molecules attached to NP and NR conjugates. Scale bar: 200 μm. The error bars correspond to the standard error of the mean (n = 3)

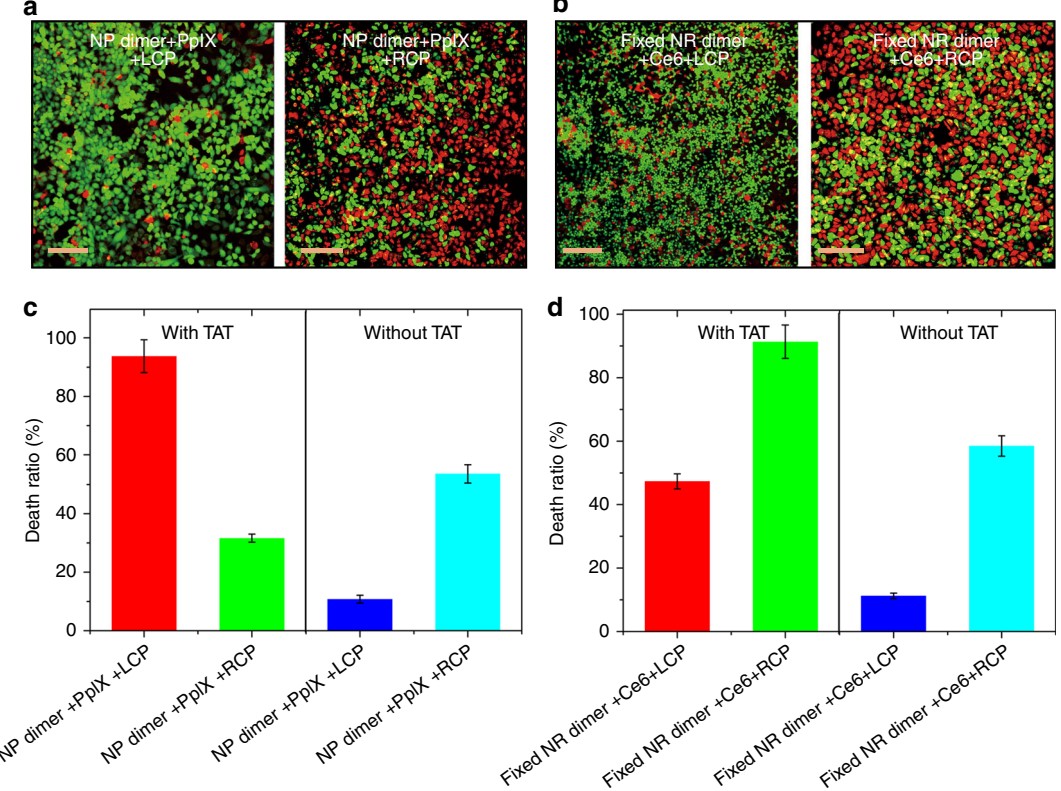

**Fig. 5** Cell penetrating peptide effect for photodynamic therapy. **a**, **b** Live (green, FITC)/dead (red, Texas Red) assays with confocal microscopy for adherent HeLa cells after 30 min illumination under different polarization conditions for NP dimers (without cell penetrating peptides labeled on the NP surface and denoted by without TAT) with 532 nm photons (**a**) and NR dimers (without cell penetrating peptides labeled on the NR surface and denoted by without TAT) with 660 nm photons (**b**), respectively. **c**, **d** The death ratio of HeLa cells incubated with (**c**) NP dimers, (**d**) NR dimers under different polarized light irradiation determined by MTT assay. The NP or NR dimers with/without cell penetrating peptides modified on the surface were denoted by with TAT/without TAT. Scale bar: 200 μm. The error bars correspond to the standard error of the mean ($n = 3$)

of singlet oxygen was estimated to be 200 ns in biological media[67]. During this time singlet oxygen can diffuse over the distance of 10–50 nm depending on the biological tissue[68, 69]. Thus, the photons absorbed inside the cells produce a much stronger effect than the photons adsorbed outside the cell boundaries, which translates in several fold increase apoptotic index for LCP vs RCP when the dimers inside the cells. The importance of the intracellular localization of the light-adsorbing species for dichroic targeting is illustrated by the comparison of efficiency of photodynamic therapy with and without cell penetrating TAT peptides that facilitate the transmembrane transport of NP and NR dimers. When TAT peptides are present on the surface of the photodynamic therapy agent, the death ratio increases for the circular polarization primarily responsible for apoptosis, that is for LCP for NP dimer (Fig. 5a,c) and RCP for NR dimer (Fig.5b,d). Since photons of opposite handedness may generate ROS both inside and outside of the cell the dependence of the death ratio for NPs/NRs carrying TAT peptides is less distinct, which reflects the multiplicity of factors that affect ROS when they are generated outside of cells before they reach the target.

## Discussion

DNA-bridged NR dimers provide an unusually sensitive readout of intra-/extracellular localization of the plasmonic particles. The chirality reversal originates in large part from the change of equilibrium conformation of the assemblies defined by interdependent electrostatic, elastic, and van der Waals interactions. The observed chiroptical effects for transmembrane NP transfer open the door to further studies of chiral nanostructures for

polarization-enhanced photodynamic therapy taking advantage of dichroic targeting of cancer cells internalizing chiroplasmonic nanostructures.

## Methods

**Cell cultures**. HeLa cells were purchased from the ATCC (American Type Culture Collection). Cells were cultured in Dulbecco's modified Eagle's medium (DMEM) supplemented with 10% fetal bovine serum (FBS) at 37 °C containing 5% $CO_2$ atmosphere.

**In vitro circular dichroism measurements**. HeLa cells were seeded in a six-well plate with a density of $10^5$ cells per well. The NP dimers (10 nM) and NR dimers @PS-PAA (5 nM) were co-cultivated with HeLa cells for 2, 8, 12, 18, 24 and 48 h at 37 °C, respectively. The cells were collected using standard trypsin-based lift-off protocol and washed with PBS three times. Then, the cells were re-dispersed in 200 μl PBS and the chiroptical activity was characterized by MOS-450/AF. The optical path length was 1 cm. The CD spectra in these experimental series were measured for cellular dispersion when the dimers, cells, and other chiroptically active species have random orientation in respect to the incident light beam. Under these conditions linear dichroism and linear birefringence of the media do not contribute to circular dichroism readout.

**Characterization**. UV–vis absorption spectra were collected using a UNICO 2100 PC spectrophotometer and processed with Origin Lab software. TEM images were obtained using a TEM JEOL JEM-2100 operating at an acceleration voltage of 200 kV. 3D tomographic images were obtained with an FEI Titan Krios equipped with a Gatan UltraScan 4000 (model 895) 16-megapixel CCD under a 300 kV electron accelerating voltage.

**Data availability**. The authors declare that all data supporting the findings of this study are available within the article and its Supplementary Information files, or are available from the authors upon request.

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

## Acknowledgements

This work is financially supported by the National Natural Science Foundation of China (21631005, 21673104, 21522102, 21503095, and 21301073). This study was supported by the Department of Army W911NF-10-1-0518 Reconfigurable Matter from Programmable Colloids. Partial support of this work was also made by the Center for Photonic and Multiscale Nanomaterials (C-PHOM) funded by the National Science Foundation (NSF) Materials Research Science and Engineering Center program DMR 1120923 as well as NSF projects 1403777; 1411014; 1463474; 1538180. P.K. acknowledges the NSF DMR-1506886 grant. The authors express appreciation to Professor Petr Kral and Mr. Cong (Scott) Wang from University of Illinois of Chicago for helpful discussions.

## Author contributions

M.S. was responsible for TEM and three-dimensional (3D) reconstruction of electron tomography. L.X. carried out NP/NR conjugation and spectroscopic measurements. J.H.B., S.A. and N.A.K. analyzed the models for dimer energies. H.K. carried out the cell viability studies. C.X. designed and planned the experiments, analyzed the results, and co-wrote the manuscript. N.A.K. conceptualized the mechanism of chirality switching. H.K. and N.A.K. analyzed the results, and co-wrote the manuscript.

## Additional information

**Competing interests:** The authors declare no competing financial interests.

