## [Peer Review File · Nature Communications]

Reviewers' comments:

Reviewer #1 (Remarks to the Author):

The paper is about a phenomenon occurring for metal nanoparticle/nanorod dimers, linked by a DNA linker, where a plasmonic CD signal appearing for the dimers dispersed in water inverts on internalization into cells.

The authors have suggested a model for the change in the CD signal. They assume that the original plasmonic CD signal is due to formation of twisted pairs of particles, where the particles are slightly elongated and claim that the change in charging of the particles is responsible for a change in the distance between the twisted pair which would lead, under very particular conditions to CD signal inversion. They also show that this effect may occur by simply changing pH or BSA concentration in the solution.

They also demonstrate an application for photodynamic therapy, where there seems to be a substantial difference in killing the cells between illumination with right hand and left handed circularly polarized light.

While I think that the observed effect of plasmonic CD inversion on insertion into cells is interesting, it is also a very complex system where the explanation of the observed effect is difficult. I think that the model proposed by the authors is a wild speculation without substantial experimental support. The usefulness of this effect is questionable, especially in real biological systems, where also in the extra-cellular matrix there are lots of chiral biomolecules which would affect the CD signal as in the case where the dimers enter the cells.

More specifically:

1. As I wrote above, the model is not convincing: the citrate based particles have random shapes and the tomography images and the angular analysis did not convince me that the original dimer CD arises from the twist. Other have shown both theoretically and experimentally that spherical particle dimers (or polymers) with a chiral molecule in between the particles would show significant plasmonic CD signal due to a "hot-spot" effect. The same would hold for nanorod pairs. This model is also sensitive to the distance between the particles and a signal inversion may also occur on changes in particle separation. Then, internalization in the cells (or interaction with BSA, for example) may cause various things, including change in particle separation. The BSA (or other cell proteins) may get into the hot-spot volume and also induce plasmonic CD signals which may be opposite to the DNA. Also slight changes in DNA conformation would cause significant change in plasmonic CD lineshape/polarity.

2. The data in figure 3 is not convincing – the encapsulation with the polymer has some effect, but not conclusive. I do not see how it helps the conclusions. It is also fair different between NPs and NRs, which cannot be explained by the authors' model.

3. The differences in cell viability with the dimers and illumination with the two different circular polarizations are too large to be explained by the small (~10 mdeg) CD signals that the dimers exhibit. Such a CD signal should be responsible for up to ~1% difference, not more, while the viability data show up to X2 effects. This seems too good to be true, at least according to the plasmonic CD mechanism.

In summary, I see several serious problems with the interpretation of the results, and have some doubts on the usefulness of the observed effect. The title itself is very misleading, I seriously doubt that the chirality of the system (i.e. dimers) really inverts on internalization in the cells, but rather a more complex and less spectacular phenomenon occurs. The paper might be considered further for publication on a significant change in the analysis of the data and conclusions and proper discussion of various possible mechanisms. One simple experiment that could be done is synthesizing metal particles that are more spherical in nature, and testing whether dimer plasmonic CD occurs or not.

Reviewer #2 (Remarks to the Author):

The investigation demonstrates that circular dichroism can be used as an in-situ tool for the internalization of chiral dimers of anisotropic gold nanoparticles, stabilized with a thiol functionalized polyethylene glycol and complementary DNA fragments. The proposed nanoparticle uptake mechanism, based on the enantiomeric reversibility, has been conveniently demonstrated by circular dichroism, electron microscopy, as well as an effective theoretical model. Additionally, the authors show that circularly polarized light can be used for selective photodynamic and photothermal therapy purposes, showing enhanced cell-induced apoptosis respect to conventional non-polarized and linear excitation. This study shows, for the first time, the potential of circular dichroism as a selective and sensitive spectroscopy to understand the interaction between gold nanoparticles and the intracellular domain, and therefore I consider its content as excellent and exciting. All methods and data seem to be of high quality and reproducible, and the conclusions are soundly-based on the results. I therefore recommend the publication of this highly relevant work on Nature Communications after some minor considerations.

- 1) Linear dichroism contributions turn up as an artefact in circular dichroism measurements under non-Brownian suspensions, as is the cell cytosol. Can the authors exclude such contribution from the plasmonic circular dichroism response of dimers within the cell? Probably, the answer can be related to the photodynamic therapy experiments, in which linear dichroism excitation was used.
- 2) Usually, the internalization of gold nanoparticles occurs via endosome formation. It is convenient that the authors explain why the plasmonic dimers are internalized directly to the cytosol at low incubation times without any endocytic process.
- 3) Anisotropic gold nanoparticles with larger aspect ratios are expected to be more efficient in terms of the plasmonic optical activity response (see *Angew. Chem. Int. Ed.* 2011, 50, 5499). Why do the authors use gold nanoparticle with short aspect ratios?
- 4) Cell killing rates under different polarized light excitations may be included in the main manuscript.

Reviewer #3 (Remarks to the Author):

Review – Xu et al, Nature Communications Submission
“Intracellular Localization of Nanoparticle Dimers by Chirality Reversal”
Sept. 2016

Summary of Review

In this manuscript, Xu et al explore the use of chiroptically-active nanoparticle dimers to study

particle intracellular localization over time. The authors use circular dichroism to detect changes in dimer conformation that result from differences in the extra- and intracellular environments that influence the conformation of DNA helices linking NPs within given dimers. A mathematical framework is developed to model the expected conformational changes in dimers as a function of energetics, and the results agree fairly well with the empirical results. The authors also demonstrate one application of chiral specificity for photothermal sensitization and therapy in cancer cells. The authors conclude that their finding “opens the door for real-time observation of transmembrane transport of plasmonic nanodrugs, quantitative assessment of nanoscale interactions, and biomedical applications of chiral nanostructures.”

In general, the manuscript is well-written and organized, and the logic behind each experiment is clear. My biggest concern is how generally translatable this work is for other nanoparticles with different surface chemistries and biomolecular targeting specificities and, perhaps most importantly, whether such constructs can be used for studies beyond cell culture (i.e., in vivo or in the presence of many extracellular components). I also believe that the manuscript can be improved, clarified, and better contextualized through the inclusion of several additional references. Having said that, I found the experiments to be designed and executed with care and attention to detail, and the results are genuinely interesting and well-explained. Most importantly, the potential application of the dimer constructs beyond simple intracellular localization studies should make this work appealing to a general audience. Therefore, this manuscript may be suitable for publication in *Nature Communications* pending an adequate address of the suggested revisions below (primarily, a small number of additional control experiments and minor revisions to the presented mathematical model). I would be happy to consider a revised version of this manuscript.

Major Comments

- The authors state that “knowing whether the nanoparticles are located inside or outside of the cell is paramount for adequate interpretation of diagnostic data and treatment efficacy....” I do not disagree, but it seems that the authors have developed an elegant yet very specific nanoparticle construct that arguably addresses only one small niche of the entire scope of nanoparticle applications in biomedicine. I would like to know if the dimer constructs can be synthesized in sufficiently high yield and uniformity (i.e., minimal monomers, trimers, etc.) so as to be useful for biomedical studies beyond cell culture. Related to the question of general applicability of the technique, one of the biggest uncertainties I have relates to extracellular exposure of NP dimers to proteins and other analytes, which I suspect might alter chiral reconfiguration or (more likely) the mechanisms by which dimers are transported into cells. The formation of a protein corona on the NPs would occur long before transmembrane passage for in vivo applications and in some instances of cell culture. The inevitable endocytosis of protein-coated dimers is a particular concern, which may limit the applicability of this technique to cell culture conditions with minimal soluble or secreted proteins. It would be useful for the authors to highlight the potential drawbacks of introducing exogenous DNA (albeit in minimal amounts) when using the dimeric constructs (this note could be included as part of the supplementary information). With these comments in mind, I recommend that the authors address potential limitations of this dimer-based intracellular localization technique in their concluding paragraphs in order to properly contextualize the study within the scope of nanomedicine. That said, I expect the experimental setup and execution will be useful for a wide host of cell culture studies.
- Lines 35-36: The authors state that “there are no methods for the conclusive determination of whether NPs are inside or outside the cell, even for cell cultures.” This claim appears to be somewhat inaccurate. Patskovsky et al have demonstrated the ability for three-dimensional NP localization in cancer cell cultures using point-spread function analysis of images acquired at multiple focal points within samples.¹ I believe this study should be cited and the sentence in question should be revised so that the authors do not undersell or neglect current capabilities in NP imaging technologies. It

should also be noted that the ability to localize NPs in 3D is also heavily dependent on the type of NP— for instance, quantum dots can be localized in 3D in a typical fluorescent confocal microscopy setup. However, I note that the present manuscript's merits extend beyond the ability to identify static particle location in cells.

- Line 138-139: The mathematical modeling of Wspr takes into account the physical properties of the dsDNA component of the NP linkers, but the model does not appear to account for the thiolated DNA anchors, which are single-stranded, poly-A sequences. The authors note that these anchors are "enshrouded by neighboring PEG and TAT molecules," although I doubt this coating would fully constrain potential torsion within the single-stranded regions. (This statement is also made in the Supplementary Information.) On a related note, there do not appear to be measurements of the PEG grafting density in the current manuscript. Thus, it is conceivable that the ssDNA regions will contribute to the extent of chiral reversal observed upon transmembrane transit. For this reason, I believe the authors should consider adding the relevant term(s) to their mathematical model to account for the ssDNA anchors. This addition may not substantially change the theoretical results (and I know the authors are presenting a minimalist model), but I believe it would provide a more rigorous framework for modeling the dimer responses. The model is perhaps not vital to the key points of the manuscript, so while it would be a less preferable revision, the authors could also keep their current model and specifically acknowledge the possible contribution of the ssDNA linkers to the extent of chiral changes – especially regarding the effect of the length of the ssDNA regions. As a related minor note: the prediction and experimental results for the extracellular energy minima match more closely (-10° vs. -9.6°) than the energy minima for the intracellular conditions ($+6^\circ$ vs. $+12.5^\circ$). This is fine, but the description of the possible forces that could cause the deviation should also include components of the dimer system not accounted for by the model (i.e., the ssDNA anchors, possible variability in the PEG layer density across the particle surface, etc.) in addition to more purely physical terms. I am surprised that hydrogen bonding was neglected in the model, especially considering its importance for dsDNA structure.

- Regarding Lines 178-180: Why was PpIX not also used for the PS-PAA-constrained NR dimers? If such a construct was not feasible, the authors should provide an explanation. I get that each different construct interacts with a unique light polarization/wavelength and that it is good to show the method's generality with multiple photosensitive agents, but the experiment suggested above should ideally be shown as a control to assay photothermal effects for cells incubated with constructs in which the conjugated photosensitizer is off-resonance with the NP/NR dimer but on-resonance with the excitation wavelength.

- Lines 191-192: The authors conclude that their model enables "extension to more complex experimental systems with biospecific interactions, and elucidation of a spectrum of media parameters on nanoscale conformations." There is no indication that the self-described "minimalist" model presented in the manuscript could accurately inform numerous interactions with soluble proteins or the response of the NP dimers upon adhesion to specific molecules of interest (or the subsequent endocytosis that may be induced upon target binding, a mechanism which would be distinct from direct translocation via the TAT peptide). Thus, I recommend that the authors seriously consider revising this statement. The model provides nice validation for the observed experimental results, but the aforementioned conclusions seem premature with respect to the model in its current form as well as complexities inherent in the behavior of different cell types in culture, etc.

- For dimers made from particles of nearly identical size and shape tethered at one locus, shouldn't there be at least some degeneracy in the ability to determine relative angles (specifically with respect to positive vs negative angles of equal magnitude), or am I forgetting something fundamental here?

- In addition to TEM images, it would be incredibly useful if the authors performed DLS measurements on each dimer construct and then presented the DLS results as size distribution histograms. Alternatively, the authors could provide manual counts of monomers, dimers, trimers, etc. from TEM images, although this approach is decidedly more tedious and would not access as large a sample size as DLS (although TEM analysis would be more accurate and sensitive).

Minor Comments

- Lines 30-31: it will be advantageous to include more recent references for applications of hyperspectral imaging, which is gaining traction as a biomedical diagnostic technique. The inclusion of additional references would strengthen and further underscore the relevance of the manuscript. Specifically, hyperspectral imaging has recently been demonstrated for quantitative biodistribution and targeted uptake studies² as well as for studying nanomaterial toxicity³.

- Line 31: the authors should include key references for the use of nanoparticles for si-RNA therapies

- Lines 39-41: the statement “Additionally, the intensity and the width of plasmonic bands is strongly affected by cellular scattering, protein adsorption, and tissue heterogeneity” should include references (there are several to choose from).

- Lines 44-45: While it is true that plasmonic NPs can quench fluorescent dyes, NPs have also been shown to actually enhance fluorescent emission. Quenching vs enhancement is heavily dependent upon a number of factors including the linker distance between the dye and NP, orientation (in the case of anisotropic NPs), and local chemical environments. In addition to citing relevant work⁴⁻⁷, the authors should revise this statement to be more accurate. It may be more accurate to say that, due to the complex quenching and enhancing interactions (radiative and non-radiative) between dyes and NPs, the results from pH-reporting dyes for NP cellular localization may be compromised.

- Line 50: It would be useful to readers if the detection sensitivity for chiroptical NPs were provided as a parenthetical note (3.7 attomolar, based on Ref 14). It would be especially useful if this detection sensitivity were provided in terms of a number of NPs instead of molarity. In other words, does chiral sensing provide single-particle detection capabilities or close to it?

- Line 51: It is arguably important to cite earlier work from the Alivisatos group on the formation of DNA-linked dimers, trimers, and lattices to acknowledge the priority of these studies with respect to references 15 and 16.

- Lines 58-59: I appreciate the authors' candor regarding their original hypothesis.

- Lines 73, 76, 80: It would be useful to briefly explain the discrepancy between the cited peak locations (likely dielectric differences in pure buffer vs cells/cell culture), especially for the shifts from 500 and 530 / 508 and 537.

- Line 88: In my opinion, “red-shifting” should be generalized to “plasmonic peak shifting” since the nature of the spectral shifts upon agglomeration are heavily dependent on nanoparticle shape (i.e., NRs often blue-shift)², surface coating (i.e., silica-coated particles can retain their spectral properties when agglomerated)², and orientation⁸.

- Lines 90-92: The Raman intensity reaches a peak near 24 hours, ostensibly due to NP agglomeration within the cells. As shown in Figure S6, the Raman intensity decreases mildly after 24 hours, which the authors suggest is possibly the result of exocytosis. The TEM image from 48 hours post-incubation appears to show some degree of NP escape from intracellular compartments, but not necessarily true

exocytosis, which is a vesicle-mediated active process. Unless the authors have TEM images that definitively show exocytosis, I would consider revising this statement. It seems that a more accurate statement would be something along the lines of “The dynamics of the Raman spectra were not found to correlate with transmembrane transport and, most likely, reflect agglomeration processes of the NP dimers followed by some degree of NP escape from intracellular compartments.”

- Line 110: As a point of interest, to what extent is the twisting motion between the NPs due to torsional strain within the dsDNA segment as opposed to the relatively unconstrained ssDNA anchors? One way to further explore this may be to use flexible ssDNA linkers of various lengths.
- Line 173: a reference should be included for the PpIX absorption band. The same is probably true for Ce6.
- The PS-PAA incubation provides a nice control experiment – is there a TEM analysis of the angle distributions of constrained vs unconstrained pairs? Alternatively, do the authors expect that a greater degree of constraint could be achieved by using a different particle coating such as a silica shell?
- Figure 1A should technically be referenced at a relevant location within the text. It is currently not mentioned.
- HeLa cells are the most straightforward choice for an initial demonstration of the dimer constructs. So while I do not consider it absolutely vital to test the particles in more than one cell type for an initial report, I wish that the authors had included at least some preliminary data for dimer uptake in neurons. Again, I am not suggesting that the authors must perform the experiment again in a different cell type, but the behavior of constructs that are sensitive to changes in dielectric environments seem like they are ideally suited for studying neuronal function and transmembrane transport, so I hope the authors explore (or are currently exploring) this avenue.
- I note that the change in CD between unconstrained and constrained dimers appears to be more significant for NPs vs. NRs (I suspect that pre-existing multiple DNA linkages between NRs may be partly responsible for this). The manuscript would be strengthened if the authors addressed possible sources for the extent of each CD change in the supplementary information.
- In Figure 5C and 5D, descriptions of the sample size should be reported, along with a description of what the error bars represent (i.e., standard error of the mean, standard deviation) in the figure legend. Units for the ROS intensity measured on the y-axis should also be included.
- Were the numbers of DNA linkers and TAT peptides per particle empirically measured as part of this study, or were they assumed from previously published synthetic methods?
- Were NRs without PS-PAA shells tested? If so, what were the key findings for the unconstrained NR dimers?

Minor Comments for Supplementary Information

- Line 18: While it is well-known, the specific citrate reduction method should be cited.
- Line 22: For the purposes of replication, the centrifugation conditions (rcf, time, etc.) should be included.
- Line 127: the word “can” should be deleted.

- Line 179: “z-potential” should be changed to “ ξ -potential” since “z” is representative of another variable in the equations.
- Regarding Figure S4: the change in intracellular NP distribution is quite abrupt between 18 and 24 hours. Can the authors explain why agglomeration appears to happen so suddenly? The same phenomenon is observed in Figures S21 and S22.
- Regarding Figure S5: panel C shows an increase of ~ 0.1 OD over the first two hours once excess NP dimers are washed out. I am curious as to why an increase of a similar magnitude is not observed in panel B. Granted that the baseline absorbance is higher when excess dimers are not washed out, shouldn't there be an increase of the amount of NP dimers within the cells during the first two hours?
- Regarding Figure S12: it may be noted elsewhere, but the sample size should be reported for the measurement of dihedral angle distribution within the figure legend.

Formatting & Clarity

- Line 33: a comma is needed to separate references 8 and 9.
- Line 34: “localization of particles with accuracies comparable to cell sizes” is slightly unclear. Perhaps “spatial accuracies” would be more clear. Alternatively, “spectroscopic localization of particles with subcellular accuracy” may be clearer wording.
- Line 63: a comma is needed to separate references 17 and 18.
- The author names listed in reference 10 are not formatted consistently with the other references (i.e., last name, first initials)
- Line 157: It is difficult to tell since the text is justified, but there may be an additional space between the words “connecting” and “two.”
- Line 171: protoporphyrin IX (PpIX) should be spelled out in addition to the abbreviation at its first mention within the text.
- An explanation of the FITC/Texas Red imaging for the live/dead cell assays should be included in the Figure 5 legend to aid the rapid interpretation of the results.

Recommended Additional References for the Manuscript

- 1 S. Patskovsky, E. Bergeron, & M. Meunier, “Hyperspectral darkfield microscopy of PEGylated gold nanoparticles targeting CD44-expressing cancer cells,” *J. Biophotonics* 8(1-2), p.162-7 (2013).
- 2 E.D. SoRelle, O. Liba, J.L. Campbell, R. Dalal, C. Zavaleta, & A. de la Zerda, “A hyperspectral method to assay the microphysiological fates of nanomaterials in histological samples,” *eLife* 2016;10.7554/eLife.16352 (2016)
- 3 G.A. Roth, S. Tahiliani, N.M. Neu-Baker, S.A. Brenner, “Hyperspectral microscopy as an analytical tool for nanomaterials,” *WIRE Nanomed. & Nanobiotechnol.* 7(4), p.565-79 (2015).
- 4 E. Dulkeith, A.C. Morteani, T. Niedereichholz, T.A. Klar, J. Feldmann, S.A. Levi, F.C.J.M. van Veggel, D.N. Reinhoudt, M. Moller, & D.I. Gittins, “Fluorescence quenching of dye molecules near gold nanoparticles: radiative and non-radiative effects,” *Phys. Rev. Lett.* 89, 203002 (2002)
- 5 G. Schneider & G. Decher, “Distance-dependent fluorescent quenching on gold nanoparticles ensheathed with layer-by-layer assembled polyelectrolytes,” *Nano Lett.* 6(3), p.530-6 (2006)
- 6 J.R. Lakowicz, “Radiative decay engineering 5: metal-enhanced fluorescence and plasmon emission,” *Anal. Biochem.* 337, p.171-94 (2005).

7 K. Aslan, M. Wu, J.R. Lakowicz, & C.D. Geddes, "Fluorescent core-shell Ag@SiO₂ nanocomposites for metal-enhanced and single nanoparticle sensing platforms," *J. Am. Chem. Soc.* 129(6), p.1524-5 (2007).

8 A.M. Funston, C. Novo, T.J. Davis, & P. Mulvaney, "Plasmon coupling of gold nanorods at short distances and in different geometries," *Nano Lett.* 9(4), p. 1651-8 (2009).

Responses to Reviewers

Reviewer #1:

Comment 1-1: *The paper is about a phenomenon occurring for metal nanoparticle/nanorod dimers, linked by a DNA linker, where a plasmonic CD signal appearing for the dimers dispersed in water inverts on internalization into cells.*

The authors have suggested a model for the change in the CD signal. They assume that the original plasmonic CD signal is due to formation of twisted pairs of particles, where the particles are slightly elongated and claim that the change in charging of the particles is responsible for a change in the distance between the twisted pair which would lead, under very particular conditions to CD signal inversion. They also show that this effect may occur by simply changing pH or BSA concentration in the solution.

They also demonstrate an application for photodynamic therapy, where there seems to be a substantial difference in killing the cells between illumination with right hand and left handed circularly polarized light.

While I think that the observed effect of plasmonic CD inversion on insertion into cells is interesting, it is also a very complex system where the explanation of the observed effect is difficult. I think that the model proposed by the authors is a wild speculation without substantial experimental support.

Reply 1-1: We agree with **Reviewer 1** in the aspect that we observed a most interesting phenomenon. We respectfully disagree, however, with the assertion that “...the model proposed by the authors is a wild speculation without substantial experimental support.” Careful examination of the content of the manuscript gives

evidence to the contrary. The *direct experimental support* to this model is provided by **Figures 1 C-H, 2 C-D, and 3 A-D**, presented in the main text of the manuscript, as well as by **Figures S1-S2, S5-S8, S11-S16, S19-S21, S23-S24, and S26-S29**, given in Supplementary Information. The experiments validating the model include different types of electron microscopy, three types of spectroscopy, and biological experiments under in-vitro conditions. These experimental data are accompanied by statistical evaluation carried out with the rigorous requirements of *Nature* journals (see **Figures 2D, S3, S14-S16, S22**). Please also note that the experimental support includes data describing *both* individual nanoparticles (**Figures S2, S5, S6, S8, S27, and S29**), and their ensembles (**Figures 1C-H, 3A-D, S1, S7, S8, S10-S13, S19-S21, S23, S24, S26, and S28**).

Besides the extensive experimental support, the concept of variable twist-angle, θ , inside and outside the cell (*i.e.* our model), is also supported by computational data (**Figures 2E, S17, and S18**) and theoretical data (**Figures 4 and S30-S33**). It is also relevant to mention that the theory utilizes experimental parameters of DNA from multiple trustworthy sources¹⁻⁴, and can also be cross-correlated with MD simulations.⁵ Therefore, we believe that our explanation of the polarity switching in chiroptical activity is meaningful and cannot be characterized as “a *wild speculation*”.

***Comment 1-2:** The usefulness of this effect is questionable, especially in real biological systems, where also in the extra-cellular matrix there are lots of chiral biomolecules which would affect the CD signal as in the case where the dimers enter the cells.*

Reply 1-2: The usefulness of the telephone was famously questioned by a very qualified person, William Orten, the president of Western Union at that time, who dismissed the idea of A. G. Bell as “...*an electrical toy*”. There are other famous examples when the argument of usefulness was dubious and then proven to be invalid. As applied to this manuscript, we respectfully disagree with **Reviewer 1** that “ ... *usefulness of this effect is questionable, especially in real biological systems ...*”. **Figure 5** shows the real biological system- the film of live cells that demonstrates the effect of circular polarization of light on the viability of the cells when gold dimers are present. The vividly observable differences of the cellular health between left and right polarization match the expectations from chirality switching.

Cell cultures are commonly used for research in cell biology and proof-of-concept studies of, for instance, drug delivery, photodynamic therapy, or biosensing with plasmonic particles. If necessary, we can provide numerous examples of such publications in the *Nature* family of journals.

From the perspective of fundamental science, we see that the goal in this manuscript is to describe this unexpected phenomenon that has not been observed before. The effect of chirality switching is rationalized in common scientific terms, which opens the road for other people to take advantage of it. Cancer treatment is only one area of knowledge where this effect is applicable, but it is certainly not limited to that. Ultimately, usefulness is a matter of economics.

Comment 1-3: 1. *As I wrote above, the model is not convincing: the citrate based particles have random shapes and the tomography images and the angular analysis did not convince me that the original dimer CD arises from the twist. Other have shown both theoretically and experimentally that spherical particle dimers (or polymers) with a chiral molecule in between the particles would show significant plasmonic CD signal due to a "hot-spot" effect. The same would hold for nanorod pairs. This model is also sensitive to the distance between the particles and a signal inversion may also occur on changes in particle separation. Then, internalization in the cells (or interaction with BSA, for example) may cause various things, including change in particle separation. The BSA (or other cell proteins) may get into the hot-spot volume and also induce plasmonic CD signals which may be opposite to the DNA. Also slight changes in DNA conformation would cause significant change in plasmonic CD lines shape/polarity.*

Reply 1-3: We understand that our model is not convincing to **Reviewer 1**. Here we shall try to explain why we had to choose the mechanism of the phenomena that is perhaps different than what he/she had in mind and what has been previously discussed in the literature.^{6,7} Let us specifically address the hot-spot model that **Reviewer 1** mentioned in the comments. It predicates that chiroptical activity in our system originates from the chiral molecules (DNA, TAT peptide, or components from the cellular milieu) placed in the hot-spots. The hot-spots enhance and spectrally shift the chiroptical activity of the biological molecules. When the molecule in the hot spot changes to, for instance, a different one, the resulting chiroplasmonic activity may switch the polarity. As we were preparing the manuscript, we did consider the hot-

spot model, but we were not able to rationally explain the experimental facts with it. Here are the reasons why we had to choose a different path for explaining chirality reversal.

1. It could be difficult to argue whether an abstract model fit a particular system or not. Fortunately, there is a well cited work by Zhang et al.⁶ that considers the hot-spot model in a NP dimer bridged by DNA (**Figure R1**). The absorption spectra of these chiral molecules and gold NPs are very similar to ours. At first glance, this is a very convenient pathway to explain our observations. However, the spectra calculated by Zhang et al.⁶ are actually quite different (**Figure R1d**) than ours (**Figures 1B** and other); they have multiple spectral features that we do not have, and vice versa. Furthermore, even when we added the molecules with different handedness and different maxima of absorption spectra, which

Figure R1. Modified Figures 1 and 2 from Ref.⁶ (a) Schematics of a model incorporating a gold dimer and a chiral molecule; (b) Extinctions of single-molecule and single noble metal NPs. (c) CD signal for the molecule–Au-dimer complex averaged over the molecular dipole orientation; the inset shows experimental data obtained in Ref.⁶³

should result in the appearance of more complex line-shapes as in **Figure 2d**

from Zhang et al.⁶ (**Figure R1**), no spectra similar to those calculated in the cited study could be observed (**Figure 1**). Therefore, it is difficult to claim the match with this model.

2. Our dimers represent a system which is far from being optimal for hot spots. Zhang et al.⁶ calculated the dependence of the hot-spot-enhanced circular dichroism peaks on the distance between the NPs. They found that it is highest when the interparticle gap is 0.5–1 nm and rapidly

Figure R2. Reproduced Figure 4 from Zhang et al.⁶. CD spectra of a Ag dimer for the molecular resonance $\lambda_0 = 400 \text{ nm}$ and for various separations, d . Inset: Small region of the spectrum.

drops as the distance increases (**Figure R2**). The gaps between the NPs in our dimers are much greater, approaching a 10x difference from the optimal one calculated by Zhang et al.⁶. At these distances, hot-spot-enhancement of polarization rotation is much less dramatic.

3. Multiple theoretical and experimental studies indicate that the intensity of the electrical field in the hot-spots is tracked by the intensity of the surface-enhanced Raman scattering,^{8,9} that is, the ‘hotter’ the hot spot, the stronger the

Raman scattering is. As shown in **Figure S8**, the intensity of Raman scattering of the band at $\sim 513\text{ cm}^{-1}$ reaches the maximum at 24 h. If the hot-spots determine the chiroptical activity in our system, the circular dichroism should also reach the maximum at 24 h, without the significant plasmonic band shift. However, this is not the case (**Figure 1E**). Therefore, the hot-spot model cannot explain this experimental fact.

4. Data in **Figure 1G** show that chiroptical activity of DNA-bridged dimer in absence of all other biological components changes the sign of polarization rotation in response to the concentration of sodium chloride, NaCl. The effect is exceptionally strong—polarization rotation switches from -25 mdeg when $[\text{NaCl}] = 0\text{ mg/mL}$ to $+30\text{ mdeg}$ when $[\text{NaCl}] = 146.3\text{ mg/mL}$. NaCl is not chiral and does not replace the DNA bridge or TAT peptides between the particles. The oligonucleotide helix retains its original handedness (shape). The change of hot-spot intensity, or orientation of DNA in it, cannot explain the sign of polarization rotation reversal in this case.
5. In our previous studies, it was shown that the presence of chiral molecules in the hot-spots between NPs or NRs is not required for observation of chiroplasmonic activity. We used *achiral* sodium carbonate to assemble the NP and NR dimers. The intensity of the chiroptical activity was comparable to that when DNA-bridges were used.¹⁰

6. Regarding the hot-spot hypothesis, we draw *Reviewer 1's* attention to the fact that the NP dimers are coated with a polyethylene glycol layer (PEG) and TAT peptides. Unlike PS-PAA shells, these layers are made prior to the NP assembly. Very floppy PEG molecules do not prevent the twisting of the NPs and are likely to even facilitate it. Furthermore, the PEG layer is fairly thick (ca. 5 nm); it is well-known that this reduces adsorption of proteins and other biomolecules onto NPs.^{11,12} Therefore, the penetration of biomolecules from the cellular milieu into the 'hot' zone is hindered by this layer.

We carried out the computational study of the hot-spot hypothesis. Chiroptical and absorbance spectra were calculated with the full space integration (4π ster) for two NP dimers, bridged by a DNA-like helix with a refractive index of 1.45 and a polarizability of $4.06 \times 10^{-23} \text{ cm}^3$. These values can be compared to those of DNA. The refractive index and polarizability of dsDNA used for bridging dimers is known to be 1.34 and $3.18 \times 10^{-23} \text{ cm}^3$, respectively.^{13,14,15} **Figure R3** presents the results of these calculations for two NP dimers. Their geometries were identical except the dihedral angle, θ , being 0 deg for one model and 13 deg for the other. The simulations reveal that, in both cases, hot spots from coupling of the plasmonic oscillations in the highly polarizable particles are indeed formed. The highest intensity of the electrical field between the NPs for the parallel dimer with $\theta = 0$ deg is 1.2 V/m and for twisted dimer with $\theta = 13$ deg is 0.6 V/m, which are values characteristic of quite "cold" hot-spots.

The maxima of the circular dichroism spectra for the parallel dimer with $\theta = 0$ deg and for the twisted dimer with $\theta = 13$ deg were 1.3 mdeg and 12 mdeg, respectively. Although hot-spots are present, their role in the chiroptical activity of the dimer is less significant ($< \sim 10x$) than the geometry of the dimer.

As per *Reviewer 1* requested, the discussion of the hot-spot mechanism has been added in the new version of the main text in pp. 6-7. The corresponding part was also added to the Supplementary Information.

7.

Figure R3 (Figure S17). Electric field (A, B) and CD spectra (C) of NP dimers bridged by DNA-like organic helix characterized by dihedral angles $\theta = 0^\circ$ and $\theta = 13^\circ$, respectively.

Comment 1-4: *The data in figure 3 is not convincing – the encapsulation with the polymer has some effect, but not conclusive. I do not see how it helps the conclusions. It is also fair different between NPs and NRs, which cannot be explained by the authors' model.*

Reply 1-4: We believe the effect of the PS-PAA shell is quite conclusive. The encapsulation with the polymer, that is, the PS-PAA shell, hinders reconfiguration of the dimers and does not allow for the chiral molecules to access the interparticle region. This finding demonstrates the essential role of reconfigurability of the dimer in the observed phenomena.

Comment 1-5: *The differences in cell viability with the dimers and illumination with the two different circular polarizations are too large to be explained by the small (~10 mdeg) CD signals that the dimers exhibit. Such a CD signal should be responsible for up to ~1% difference, not more, while the viability data show up to X2 effects. This seems too good to be true, at least according to the plasmonic CD mechanism.*

Reply 1-5: We appreciate this this comment and, indeed, the effect might appear to be much stronger than expected. This why it could be useful (**Comment 1-2**). The dimer coupled with PpIX kills cancer cells with a death ratio between LCP and RCP that is about 1.4-fold (**Figure 5A,C and S40A**). The reason for such remarkable enhancement of cell death with the change of circular polarization of the incident light is not how much light is adsorbed. It is related to WHERE it is adsorbed. Induction of apoptosis by generating reactive oxygen species INSIDE THE CELL is more

effective than outside of the cell in killing the cancer cells.

Comment 1-6: *In summary, I see several serious problems with the interpretation of the results, and have some doubts on the usefulness of the observed effect. The title itself is very misleading, I seriously doubt that the chirality of the system (i.e. dimers) really inverts on internalization in the cells, but rather a more complex and less spectacular phenomenon occurs. The paper might be considered further for publication on a significant change in the analysis of the data and conclusions and proper discussion of various possible mechanisms. One simple experiment that could be done is synthesizing metal particles that are more spherical in nature, and testing whether dimer plasmonic CD occurs or not.*

Reply 1-6: We appreciate the critique and hopefully the replies above clarified our point of view. As recommended, we carried out the experiment with the homodimers assembled by two identical sized NPs with the same dsDNA strand used in the manuscript and evaluated the chiroptical activity of these assemblies. As shown in the TEM images (**Figure R4A**), the geometries of the new 10 nm NPs are nearly ideal spheres. The aspect ratio of the nanoparticle was calculated to be 1.05 on the basis of the statistical analysis from the TEM images. The assembled dimers produce only very weak CD signals in the visible region (**Figure R4B**). Furthermore, the conjugated and elongated NPs used in the manuscript justly showed very weak chiroptical activities.

Figure R4 | TEM images (A), UV-Vis and CD spectra (B) of 10 nm NP dimers.

Reviewer #2:

Comment 2-1: The investigation demonstrates that circular dichroism can be used as an in-situ tool for the internalization of chiral dimers of anisotropic gold nanoparticles, stabilized with a thiol functionalized polyethylene glycol and complementary DNA fragments. The proposed nanoparticle uptake mechanism, based on the enantiomeric reversibility, has been conveniently demonstrated by circular dichroism, electron microscopy, as well as an effective theoretical model. Additionally, the authors show that circularly polarized light can be used for selective photodynamic and photothermal therapy purposes, showing enhanced cell-induced apoptosis respect to conventional non-polarized and linear excitation. This study shows, for the first time, the potential of circular dichroism as a selective and sensitive spectroscopy to understand the interaction between gold nanoparticles and the intracellular domain, and therefore I consider its content as excellent and exciting. All methods and data seem to be of high quality and reproducible, and the conclusions are soundly-based on the results. I therefore recommend the publication of this highly relevant work on Nature

Communications after some minor considerations.

Linear dichroism contributions turn up as an artefact in circular dichroism measurements under non-Brownian suspensions, as is the cell cytosol. Can the authors exclude such contribution from the plasmonic circular dichroism response of dimers within the cell? Probably, the answer can be related to the photodynamic therapy experiments, in which linear dichroism excitation was used.

Reply 2-1: Thank you very much for your comments. We also very much appreciate the suggestion about linear dichroism (LD) in the cytosol. The experiments addressing this comment were added to the manuscript.

LD spectra of the cells with, and without, NP dimers were measured after 8 h of incubation. As one can see in **Figure S9**, NP dimers incubated with cells displayed weak LD in the visible spectrum. We also found that there was no clear LD peak for cells alone in the visible and NIR ranges of the spectrum that could interfere with the CD spectra. Similarly, LD of cell cytosol is unlikely to affect the photodynamic therapy efficiency for left- and right-circularly polarized light.

Figure R5 (Figure S9). Linear dichroism (A) and UV-Vis absorbance (B) spectrum of the individual cells and NP dimers incubating with cells after 8 h onto the glass.

Comments 2-2: Usually, the internalization of gold nanoparticles occurs via endosome formation. It is convenient that the authors explain why the plasmonic dimers are internalized directly to the cytosol at low incubation times without any endocytic process.

Reply 2-2: This is correct. We also agree that endocytosis is better to avoid for photodynamic therapy (PDT) because the radicals are most efficient when they are generated in cytosol. The direct through-membrane transport of the particles is due to the use of cell-penetrating TAT peptides attached to the surface of NP dimers.^{16,17}

To demonstrate the importance of using TAT peptides in our case, we have incubated the cell with NP dimers which were only modified with PEG molecules. UV-Vis spectra showed no plasmonic peak at 525 nm, indicating that the NP dimers were not able to penetrate the cell membrane, even after 48 h of incubation (**Figure R6**).

Figure R6 (Figure S13). The UV-Vis spectra of NP dimers without cell-penetrating peptide modification, incubated with cells for 48 h.

Comment 2-3 *Anisotropic gold nanoparticles with larger aspect ratios are expected to be more efficient in terms of the plasmonic optical activity response (see Angew. Chem. Int. Ed. 2011, 50, 5499). Why do the authors use gold nanoparticle with short aspect ratios?*

Reply 2-3: We strongly agree with this point. The question about the NRs is also a very valid one. Au NRs will produce angled dimers with greater asymmetry¹⁸⁻²⁰. We tested NR dimers modified with PEG molecules and cell-penetrating peptides for different incubation times with HeLa cells. However, the NR dimers are not stable enough in the cytosol. From the bio-TEM in **Figure R7**, one can see that the NR dimers in cell were aggregated in the cytosol and the cetyltrimethylammonium bromide (CTAB) bilayer residues onto the surface of NRs

Figure R7 (Figure S23). The bio-TEM image of NR dimers without PS-PAA coating after being incubated with HeLa cells for 8 h.

In order to make NR dimers stable in cytosol, we had to coat the NR dimer with dense PS-PAA polymer, but then they lost reconfigurability. We have added the related results and references into the Supplementary Information.

Comment 2-4: *Cell killing rates under different polarized light excitations may be included in the main manuscript.*

Reply 2-4: Thank you very much for your suggestion about including the cell killing rates under different polarizations of the incident light. We have added the statistics data of cell killing rates under different polarized light excitations according to the grey value of confocal images. As shown in **Figure R8**, the maximum death ratio for unconstrained NP pairs and constrained NR pairs could be $93.5 \pm 3.5\%$ and $94.5 \pm 2.5\%$ under the appropriate 532 nm LCP light and 660 nm RCP light, respectively. Furthermore, LCP light gives a three-fold higher death ratio in cancer cells compared with RCP light or LP light for unconstrained NP pairs, and the similar experiment for constrained NRs could show that RCP light gives a higher percentage of apoptotic cells than LCP due to its stronger absorption by NR dimers. These results could further confirm the conclusion of the main text. We added the related results and comments into the Supplementary Information as follows:

Figure R8 (Figure S40). The death ratio of HeLa cells incubated with (A) NP dimer, (B) NR dimer under different polarized light excitations.

Reviewer #3:

Comment 3-1: *In this manuscript, Xu et al explore the use of chiroptically-activity nanoparticle dimers to study particle intracellular localization over time. The authors use circular dichroism to detect changes in dimer conformation that result from differences in the extra- and intracellular environments that influence the conformation of DNA helices linking NPs within given dimers. A mathematical framework is developed to model the expected conformational changes in dimers as a function of energetics, and the results agree fairly well with the empirical results. The authors also demonstrate one application of chiral specificity for photothermal sensitization and therapy in cancer cells. The authors conclude that their finding “opens the door for real-time observation of transmembrane transport of plasmonic nanodrugs, quantitative assessment of nanoscale interactions, and biomedical applications of*

chiral nanostructures.”

In general, the manuscript is well-written and organized, and the logic behind each experiment is clear. My biggest concern is how generally translatable this work is for other nanoparticles with different surface chemistries and biomolecular targeting specificities and, perhaps most importantly, whether such constructs can be used for studies beyond cell culture (i.e., in vivo or in the presence of many extracellular components). I also believe that the manuscript can be improved, clarified, and better contextualized through the inclusion of several additional references. Having said that, I found the experiments to be designed and executed with care and attention to detail, and the results are genuinely interesting and well-explained. Most importantly, the potential application of the dimer constructs beyond simple intracellular localization studies should make this work appealing to a general audience. Therefore, this manuscript may be suitable for publication in Nature Communications pending an adequate address of the suggested revisions below (primarily, a small number of additional control experiments and minor revisions to the presented mathematical model). I would be happy to consider a revised version of this manuscript.

Reply 3-1: Thank you very much for your constructive comments, suggested revisions, and critical points.

Comment 3-2: 1. *The authors state that “knowing whether the nanoparticles are located inside or outside of the cell is paramount for adequate interpretation of diagnostic data and treatment efficacy....” I do not disagree, but it seems that the*

authors have developed an elegant yet very specific nanoparticle construct that arguably addresses only one small niche of the entire scope of nanoparticle applications in biomedicine. I would like to know if the dimer constructs can be synthesized in sufficiently high yield and uniformity (i.e., minimal monomers, trimers, etc.) so as to be useful for biomedical studies beyond cell culture. Related to the question of general applicability of the technique, one of the biggest uncertainties I have relates to extracellular exposure of NP dimers to proteins and other analytes, which I suspect might alter chiral reconfiguration or (more likely) the mechanisms by which dimers are transported into cells. The formation of a protein corona on the NPs would occur long before transmembrane passage for in vivo applications and in some instances of cell culture. The inevitable endocytosis of protein-coated dimers is a particular concern, which may limit the applicability of this technique to cell culture conditions with minimal soluble or secreted proteins. It would be useful for the authors to highlight the potential drawbacks of introducing exogenous DNA (albeit in minimal amounts) when using the dimeric constructs (this note could be included as part of the supplementary information). With these comments in mind, I recommend that the authors address potential limitations of this dimer-based intracellular localization technique in their concluding paragraphs in order to properly contextualize the study within the scope of nanomedicine. That said, I expect the experimental setup and execution will be useful for a wide host of cell culture studies.

Reply 3-2: We appreciate the points raised by **Reviewer 3** about difficulties in the transition from cell culture to clinical practice and agree that claiming the immediate applicability of these constructs is premature. However, we also think that knowing whether the nanostructure is located inside or outside of the cell is essential for many applications of plasmonic and other nanoparticles, both diagnostic and therapeutic, currently being discussed in the literature.²¹⁻²³ From this standpoint, we respectfully argue that chirality switching is something better than “*one small niche of the entire scope of nanoparticle applications in biomedicine*” because it adds a new capability for the detection of intracellular/extracellular localization of nanoparticles. As one can see from Figure 5, it makes a big difference to generate the radical in the right place.

We agree that making exotic constructs in miniscule yield would not be a strong case for their practicality. Although there is a long way to translation, especially for cancer treatment, we are happy to state that the yield of NP dimers is high, reaching $85 \pm 4.2\%$. We have added these data describing the composition of the dimerization reaction to this version of the manuscript in **Figure S3**.

Figure R9 (Figure S3). Statistical analysis of different products in the reactions of dimer assembly.

The comment about protein corona is valid. Coating NPs with PEG-5000 soft

shell reduces protein adsorption but does not bypass corona formation.²⁴⁻²⁷ In fact, corona formation is desirable for the RES stealth and successful delivery of the dimers to potential targets.^{28,29} The formation of corona is welcome in our case, and a comment about it was added to the main text of the manuscript in pp. 3-4.

Endocytosis was a concern for us as well. For that reason we used TAT peptides derived from HIV-virus; these peptides promote intracellular particle transport by direct fusion of PEG layer with cellular membrane.^{30,31} Addition of TAT peptides enables the NP dimers to cross the membrane and penetrate directly into cytosol,^{30,32} and therefore avoid endocytosis and endosomal segregation.

Clarification about endocytosis was added in p. 4. The new data about transport of NPs without TAT peptide as a negative control for the study was also added as **Figure S13 (Figure R6)**.

The safety of exogenous DNA does not appear to be problematic to our systems because (a) the sequence of DNA was a non-coding region (checked from NIH genetic sequence database) and could hardly alter cell behavior, and (b) NPs prevent penetration of DNA into cellular nuclei or mitochondria where it can be integrated with the centromeres or sub-telomeric regions to cause mutation.³³ The comment about the safety concerns of exogenous DNA was added into the SI after **Figure S3**.

Comment 3-3: 2. Lines 35-36: The authors state that “there are no methods for the

conclusive determination of whether NPs are inside or outside the cell, even for cell cultures.” This claim appears to be somewhat inaccurate. Patskovsky et al have demonstrated the ability for three-dimensional NP localization in cancer cell cultures using point-spread function analysis of images acquired at multiple focal points within samples. I believe this study should be cited and the sentence in question should be revised so that the authors do not undersell or neglect current capabilities in NP imaging technologies. It should also be noted that the ability to localize NPs in 3D is also heavily dependent on the type of NP—for instance, quantum dots can be localized in 3D in a typical fluorescent confocal microscopy setup. However, I note that the present manuscript’s merits extend beyond the ability to identify static particle location in cells.

Reply 3-3: We agree with this comment and corrected the omission in the new version. Hyperspectral methods similar to those used by Patskovsky et al. were not elaborated in the first version of the manuscript because they answer the question “*How to accurately localize a single particle in space?*”, which is, in our opinion, different than the questions we posed. We agree that they are more relevant than we originally thought and we gladly incorporated a comment about their merits in p. 3 of the new version of the manuscript.

In brief, the hyperspectral methods based on deconvolution of point-spread functions are powerful. The difference between hyperspectral and chiroplasmonic methods is that the former identifies the geometrical position of the particles and the cell membrane and then determines the localization of the particle with respect to the

interior of the cells, while the chiroplasmonic method allows one to determine the localization directly from the spectra. The advantage of the hyperspectral imaging is accuracy. The advantage of the chiroplasmonic method is simplicity and return of ensemble-averaged data.

Comment 3-4: *Line 138-139: The mathematical modeling of Wspr takes into account the physical properties of the dsDNA component of the NP linkers, but the model does not appear to account for the thiolated DNA anchors, which are single-stranded, poly-A sequences. The authors note that these anchors are “enshrouded by neighboring PEG and TAT molecules,” although I doubt this coating would fully constrain potential torsion within the single-stranded regions. (This statement is also made in the Supplementary Information.) On a related note, there do not appear to be measurements of the PEG grafting density in the current manuscript. Thus, it is conceivable that the ssDNA regions will contribute to the extent of chiral reversal observed upon transmembrane transit. For this reason, I believe the authors should consider adding the relevant term(s) to their mathematical model to account for the ssDNA anchors. This addition may not substantially change the theoretical results (and I know the authors are presenting a minimalist model), but I believe it would provide a more rigorous framework for modeling the dimer responses. The model is perhaps not vital to the key points of the manuscript, so while it would be a less preferable revision, the authors could also keep their current model and specifically acknowledge the possible contribution of the ssDNA linkers to the extent of chiral changes – especially*

regarding the effect of the length of the ssDNA regions. As a related minor note: the prediction and experimental results for the extracellular energy minima match more closely (-10° vs. -9.6°) than the energy minima for the intracellular conditions ($+6^\circ$ vs. $+12.5^\circ$). This is fine, but the description of the possible forces that could cause the deviation should also include components of the dimer system not accounted for by the model (i.e., the ssDNA anchors, possible variability in the PEG layer density across the particle surface, etc.) in addition to more purely physical terms. I am surprised that hydrogen bonding was neglected in the model, especially considering its importance for dsDNA structure.

Reply 3-4: These are astute observations and we are glad that we have a chance to comment on them because anchors represent the chemistry part behind the physics and math part of **Figures 4, S30-S33**. We chose the thiolated DNA anchors containing 10 adenines as a part of the dsDNA bridge because methylamine groups in adenines are strongly adsorbed onto the surface of NPs. Adenine-gold intermolecular interactions increase the robustness of the construct and transfer of torque. In perspective of the NP-NP dimer geometry, this segment does not extend away from the surface and does not contribute appreciably to the bridge length. We added a specific comment about its role in calculations to the main text in p. 9.

Hydrogen bonding energy in DNA also changes with the torsion angle. We used the, so called, Density Functional Theory (DFT) calculations with basis set superposition error

(BSSE) correction to assess the effect of twisting on hydrogen bonding energy in DNA on the total energy of the dimer as it depends on the dihedral angle θ .³⁴ The DFT+BSSE method was used for the calculation of the energy of hydrogen bonding of dsDNA inside and outside of the cell with parameters given in **Table S1**.³⁴ The structure of dsDNA base pairs were calculated at the atomic level (**Figure S32**). The energy of hydrogen bonds was found to be ca. 2 to 5 x 10⁻²² J (**Figure S33**), which is considerably smaller than the other contributions (**Figure 4**).

The new **Figure S33** and the corresponding comment were added to SI.

Table S1 Atomic charges of nucleic acid bases used in the calculation.

Adenine			Thymine			Guanine			Cytosine		
	Extrac ellular	Intrace llular		Extrac ellular	Intrace llular		Extrac ellular	Intrace llular		Extrac ellular	Intrace llular
H	0.36	0.24	H	0.32	0.22	H	0.36	0.32	H	0.31	0.29
N9	-0.45	-0.37	N1	-0.32	-0.18	N9	-0.39	-0.21	N1	-0.38	-0.36
C8	0.15	0.11	C6	-0.18	-0.14	C8	0.15	0.08	C6	0.01	0.01
H8	0.17	0.14	H6	0.21	0.20	H8	0.16	0.11	H6	0.20	0.18
N7	-0.55	-0.36	C5	0.04	0.01	N7	-0.56	-0.42	C5	-0.46	-0.32
C5	0.01	0.02	C7	-0.36	-0.32	C5	0.26	0.11	H5	0.18	0.14
C6	0.68	0.54	H71	0.11	0.07	C6	0.44	0.32	C4	0.79	0.71
N6	-0.87	-0.71	H72	0.11	0.07	O6	-0.53	-0.42	N4	-0.96	-0.86
H61	0.40	0.28	H73	0.11	0.08	N1	-0.45	-0.41	H41	0.43	0.32
H62	0.40	0.31	C4	0.56	0.41	H1	0.35	0.25	H42	0.43	0.32
N1	-0.75	-0.63	O4	-0.55	-0.42	C2	0.60	0.75	N3	-0.73	-0.77
C2	0.54	0.42	N3	-0.39	-0.31	N2	-0.86	-0.25	C2	0.83	0.73
H2	0.07	0.06	H3	0.31	0.24	H21	0.40	0.28	O2	-0.64	-0.41
N3	-0.76	-0.53	C2	0.57	0.52	H22	0.40	0.28			
C4	0.59	0.52	O2	-0.57	-0.43	N3	-0.58	-0.43			
						C4	0.27	0.18			

Figure R10 (Figure S32) Apex (**A**) and side (**B**) views of the atomic DFT model of dsDNA used in the DFT calculation.

Figure R11 (Figure S33). Calculation of hydrogen bonding energy according to the BSSE formalism³⁴.

Regarding the comparison between calculated and experimental results for the energy minima in extracellular (-10° vs. -9.6°) and intracellular conditions ($+6^\circ$ vs. $+12.5^\circ$), we believe that the theoretical predictions for the energy minima for intracellular conditions are less accurate because of the greater heterogeneity of the dielectric constant value in the intracellular compared to the extracellular environment. A comment about it was added to p. 9 of the main text.

Comment 3-5: *Regarding Lines 178-180: Why was PpIX not also used for the PS-PAA-constrained NR dimers? If such a construct was not feasible, the authors should provide an explanation. I get that each different construct interacts with a unique light polarization/wavelength and that it is good to show the method's generality with multiple photosensitive agents, but the experiment suggested above should ideally be shown as a control to assay photothermal effects for cells incubated with constructs in*

which the conjugated photosensitizer is off-resonance with the NP/NR dimer but on-resonance with the excitation wavelength.

Reply 3-5: It is possible to conjugate PpIX to PS-PAA-constrained NR dimers. Per **Reviewer 3's** request, we evaluated the photodynamic cell killing rate for PS-PAA-constrained (fixed) NR dimers conjugated with PpIX with 660 nm laser illumination as a control experiment. Similarly to **Figure S34**, the cells incubated without NR dimers revealed >98% live cells after irradiation with 660 nm laser, 5 mW/cm². A small amount of cell death could be observed in a cell culture treated with PS-PAA-constrained NR dimers after 660 nm RCP (**Figure R12, S38**). This cellular death was much smaller than for dimers made from NPs and PpIX (**Figure 5A, S37, top left**) because NRs and PpIX have almost no spectral overlap for 660 nm photons because the absorption band of PpIX at 630 nm is weak (**Figure R13**). Note that circular polarization for incident light was chosen to maximize the adsorption inside the cells.

Figure S38. The fluorescence microscopy images for the live/dead assays for HeLa cells after 30 min irradiation under (A) right-handed circular polarized (RCP) light for PS-PAA-constrained NR dimers conjugated with PpIX and (B) linearly polarized light

(LP) for free PpIX with 660 nm photons (5 mW/cm²).

Figure R13 | UV-Vis spectra of free PpIX, Ce6, NP dimers, and NR dimers.

Comment 3-6: Lines 191-192: *The authors conclude that their model enables “extension to more complex experimental systems with biospecific interactions, and elucidation of a spectrum of media parameters on nanoscale conformations.” There is no indication that the self-described “minimalist” model presented in the manuscript could accurately inform numerous interactions with soluble proteins or the response of the NP dimers upon adhesion to specific molecules of interest (or the subsequent endocytosis that may be induced upon target binding, a mechanism which would be distinct from direct translocation via the TAT peptide). Thus, I recommend that the authors seriously consider revising this statement. The model provides nice validation for the observed experimental results, but the aforementioned conclusions seem premature with respect to the model in its current form as well as complexities inherent in the behavior of different cell types in culture, etc.*

Reply 3-6: The reason why we formulated the conclusions in this way is that the energies of DNA-DNA and many antigen-antibody interactions are known. Nevertheless, we agree that the minimalist model must fail under some conditions and since we do not know yet what these conditions are, this sentence should be revised. The new concluding statement is “*This model enables quantization of the reconfiguration process for NP/NR dimers and elucidation of a spectrum of media parameters on nanoscale conformations.*”

Comment 3-7: *For dimers made from particles of nearly identical size and shape tethered at one locus, shouldn't there be at least some degeneracy in the ability to determine relative angles (specifically with respect to positive vs negative angles of equal magnitude), or am I forgetting something fundamental here?*

Reply 3-7: This is indeed an important point. In the first approximation (*i.e.* **Figure 4**) there is *no degeneracy* between the positive and negative angles because it was removed by the torsional deformation of the spring, (*i.e.* DNA). There is, of course, the degeneracy of θ in the geometrical sense because the ends of the ellipsoid during the formation of the dimer can be flip-flopped, which may be treated as $\theta \pm 180, \pm 360 \dots$ deg. Some “true” degeneracies might arise for large deformational energies and angles of partial compensation of different interactions and collective effects in the PEG layer.

Comment 3-8. *In addition to TEM images, it would be incredibly useful if the authors performed DLS measurements on each dimer construct and then presented the DLS*

results as size distribution histograms. Alternatively, the authors could provide manual counts of monomers, dimers, trimers, etc. from TEM images, although this approach is decidedly more tedious and would not access as large a sample size as DLS (although TEM analysis would be more accurate and sensitive).

Reply 3-8: The dynamic light scattering (DLS) results of the self-assembled NP dimers were added to this version of the manuscript. The hydrodynamic diameters of the NPs were 22 ± 3 nm, and after assembly, the diameters of the dimers increased to 65 ± 2 nm (**Figure R14**). These values match very well with the expected sizes of NP pairs.

The yield of NP dimers was calculated in the ‘tedious’ way using TEM images; it was $85 \pm 4.2\%$ (**Figure S3**).

Figure R14 (Figure S2) Figure S2. Dynamic light scattering (DLS) spectra of the NPs, NP-DNA conjugate, and DNA-bridged NP dimers.

Comment 3-9: 8. Lines 30-31: *it will be advantageous to include more recent references for applications of hyperspectral imaging, which is gaining traction as a biomedical diagnostic technique. The inclusion of additional references would strengthen and further underscore the relevance of the manuscript. Specifically, hyperspectral imaging has recently been demonstrated for quantitative bio distribution and targeted uptake studies² as well as for studying nanomaterial toxicity³.*

Reply 3-9: Thank you very much for this suggestion. We have added the additional references for applications of hyperspectral imaging in p. 1 and p. 2 of the main text as follows: *“Plasmonic nanoparticles (NPs) and nanorods (NRs) are currently the basis of several potent medical diagnostic techniques and treatment options, as exemplified by photodynamic/photothermal therapy,³⁵⁻³⁸ gene and protein delivery,^{39,40} hyperspectral imaging,^{41,42,43,44,45} and siRNA-based medicine.^{46,47}”*

And then: *“The hyperspectral methods of particle localization are powerful.^{41,44,48,49,50,51} They identify the geometrical position of the single particles in respect to the cell membrane⁵² or intracellular compartments.⁴¹ The nanometer-scale accuracy of point-spread functions deconvolution allows one to determine NP localization in respect to the interior of the single cells. Methods of rapid and accurate assessment of particle localization applicable not only to single cells but to their ensembles, are also needed for diagnostics and treatment strategies based on different plasmonic effects.”*

Comment 3-10. Line 31: *the authors should include key references for the use of*

nanoparticles for si-RNA therapies

Reply 3-10: Done.

Comment 3-11: *Lines 39-41: the statement “Additionally, the intensity and the width of plasmonic bands is strongly affected by cellular scattering, protein adsorption, and tissue heterogeneity” should include references (there are several to choose from).*

Reply 3-11: There are indeed numerous references regarding these effects. We added the ones that, in our opinion, are essential for the understanding of the problems in the field.

Comment 3-12: *Lines 44-45: While it is true that plasmonic NPs can quench fluorescent dyes, NPs have also been shown to actually enhance fluorescent emission. Quenching vs enhancement is heavily dependent upon a number of factors including the linker distance between the dye and NP, orientation (in the case of anisotropic NPs), and local chemical environments. In addition to citing relevant work⁴⁻⁷, the authors should revise this statement to be more accurate. It may be more accurate to say that, due to the complex quenching and enhancing interactions (radiative and non-radiative) between dyes and NPs, the results from pH-reporting dyes for NP cellular localization may be compromised.*

Reply 3-12: This is correct that plasmonic NPs can both quench and enhance the luminescence, depending on the distance and the configuration of exciton-plasmon hybridized states.⁵³⁻⁵⁵ We highlighted quenching in the previous version because this

effect is more detrimental for the particle localization.

We agree with the corrections and added the point about the complexity of the optical phenomena involving excitonic and plasmonic states. The corresponding section of the new version of the manuscript now reads: *”Both quenching and enhancement of luminescence can result from exciton-plasmon resonance between dyes and NPs,^{23–25} which makes emission of NP conjugates with cell-reporting dyes difficult to interpret. Cellular auto-fluorescence in the visible spectrum and photo-bleaching of fluorescent dyes further complicate the task.”*

Comments 3-13: *Line 50: It would be useful to readers if the detection sensitivity for chiroptical NPs were provided as a parenthetical note (3.7 attomolar, based on Ref 14). It would be especially useful if this detection sensitivity were provided in terms of a number of NPs instead of molarity. In other words, does chiral sensing provide single-particle detection capabilities or close to it?*

Reply 3-13: This particular method does not provide single-dimer sensitivity. Under conditions of the study cited by **Reviewer 3** (i.e. Ma, W. et al.), attomolar DNA detection with chiral nanorod assemblies, *Nat. Commun.* **4**, 2689 (2013), correspond to the detection of 1100 assembled nanorods. A variant of the chiroplasmonic technique utilizing core-shell system is capable of zeptomolar sensitivity for biological detection, which is approaching the level of a few plasmonic particles in the sample volume.⁵⁶ In principle, the techniques based on NP synthesis/assembly combined with post-

processing steps afford exceptionally high levels of sensitivity.⁵⁷

Comment 3-14. *Line 51: It is arguably important to cite earlier work from the Alivisatos group on the formation of DNA-linked dimers, trimers, and lattices to acknowledge the priority of these studies with respect to references 15 and 16.*

Reply 3-14: Agree. We are happy to cite the work from the Alivisatos group.

Comment 3-15: *Lines 58-59: I appreciate the authors' candor regarding their original hypothesis.*

Reply 3-15: We went through several hypothesis of the proposed phenomenon. Some others are addressed in response to **Reviewer 1**.

Comment 3-16: *Lines 73, 76, 80: It would be useful to briefly explain the discrepancy between the cited peak locations (likely dielectric differences in pure buffer vs cells/cell culture), especially for the shifts from 500 and 530 / 508 and 537.*

Reply 3-16: We very much appreciate the thoroughness of **Reviewer 3**. We added to p. 5 the point that “*The shift of the chiroptical extrema is ascribed to the differences of NP dielectric micro-environments of the dimers inside and outside of the cells*”.

Comment 3-17: *Line 88: In my opinion, “red-shifting” should be generalized to “plasmonic peak shifting” since the nature of the spectral shifts upon agglomeration*

are heavily dependent on nanoparticle shape (i.e., NRs often blue-shift)², surface coating (i.e., silica-coated particles can retain their spectral properties when agglomerated)², and orientation⁸.

Reply 3-17: We agree with the comment made by **Reviewer 3** about the nature of the spectral shifts. The corresponding section in p. 5 about the relationship between NP agglomeration and spectral shifts is as follows: “*Importantly, the UV-Vis absorption peaks of NPs under the same conditions displayed little change (Figure S7). Despite the fact that NP agglomeration inside the cells is conducive to the shift of UV-Vis plasmon peaks due to alterations in dielectric conditions around the clusters^{43,58} broadening of the peaks makes it difficult to register them accurately.*”

Comment 3-18: *Lines 90-92: The Raman intensity reaches a peak near 24 hours, ostensibly due to NP agglomeration within the cells. As shown in Figure S6, the Raman intensity decreases mildly after 24 hours, which the authors suggest is possibly the result of exocytosis. The TEM image from 48 hours post-incubation appears to show some degree of NP escape from intracellular compartments, but not necessarily true exocytosis, which is a vesicle-mediated active process. Unless the authors have TEM images that definitively show exocytosis, I would consider revising this statement. It seems that a more accurate statement would be something along the lines of “The dynamics of the Raman spectra were not found to correlate with transmembrane transport and, most likely, reflect agglomeration processes of the NP dimers followed by some degree of NP escape from intracellular compartments.”*

Reply 3-18: We agree that it is better to err on the side of caution, and since the NP may escape the cells via exocytosis and other mechanisms, which is also cell-dependent, we have revised the statements in the main text as suggested by **Reviewer 3**. This section now reads: “*..dynamics of the Raman spectra does not correlate with transmembrane transport; it reflects formation of plasmonic hot-spots in the multi-particle agglomerates and their subsequent escape from intracellular compartments.*”

Comment 3-19: *Line 110: As a point of interest, to what extent is the twisting motion between the NPs due to torsional strain within the dsDNA segment as opposed to the relatively unconstrained ssDNA anchors? One way to further explore this may be to use flexible ssDNA linkers of various lengths.*

Reply 3-19: We believe that anchors represented by the thiols and adenine segments are strongly bound to the surface. So they are restricted in motion. Their translation or rotation would result in irreversible change of polarity of plasmonic ‘wave’. Therefore, we believe that all the deformations are concentrated in the dsDNA segment. We agree with **Reviewer 3** that it is possible to explore it further with different linkers and arrive to the dimer that exhibits such irreversibility.

Comment 3-20: *Line 173: a reference should be included for the PpIX absorption band. The same is probably true for Ce6.*

Reply 3-120: We agree and added the references for the PpIX and Ce6 photosensitizers as follows. *Podbielska, H. et al. Silica sol-gel matrix doped with*

Photolon molecules for sensing and medical therapy purposes. Biomol. Eng. 24, 425–433 (2007); Liu, K. et al. Simple Peptide-Tuned Self-Assembly of Photosensitizers towards Anticancer Photodynamic Therapy. Angew. Chemie - Int. Ed. 55, 3036–3039 (2016).

Comment 3-21: *The PS-PAA incubation provides a nice control experiment – is there a TEM analysis of the angle distributions of constrained vs unconstrained pairs? Alternatively, do the authors expect that a greater degree of constraint could be achieved by using a different particle coating such as a silica shell?*

Reply 3-21: We also think that this is an informative experiment. Analyses of the dihedral angles of the NP dimers after coating the PS-PAA are now given in **Figure S22/R15**. The average dihedral angles in cell cultures are -9.6 ± 1 mdeg and -10.4 ± 1.3 mdeg before and after PS-PAA encapsulation, respectively. The average dihedral angle increased slightly after coating PS-PAA, although not exceeding experimental error (potentially due to electrostatic charge on the PS-PAA shells).

Figure R15 (Figure S22). Statistical analysis of the dihedral angles for NP dimers in cell culture before (A) and after (B) encapsulation by PS-PAA.

Silica shells would be a good idea, indeed. They would provide a firmer coating around the dimer. This idea crossed my mind and it is worth pursuing to produce environmentally robust chiroptical constructs with the possibilities of multiple subsequent modifications⁵⁹ and long shelf life. However, deposition of SiO₂ shells on NP constructs need a certain level of experience to avoid uncontrolled coagulation of NPs. We do not have students with such experience at the moment.

Comment 3-22: *Figure 1A should be technically referenced at a relevant location within the text. It is currently not mentioned.*

Reply 3-22: Thank you for for the careful reading of the manuscript. **Figure 1A** is now referenced in p.3 in the new version of the manuscript.

Comment 3-23: *HeLa cells are the most straightforward choice for an initial demonstration of the dimer constructs. So while I do not consider it absolutely vital to test the particles in more than one cell type for an initial report, I wish that the authors had included at least some preliminary data for dimer uptake in neurons. Again, I am not suggesting that the authors must perform the experiment again in a different cell type, but the behavior of constructs that are sensitive to changes in dielectric environments seem like the are ideally suited for studying neuronal function and transmembrane transport, so I hope the authors explore (or are currently exploring) this avenue.*

Reply 3-23: We sincerely appreciated **Reviewer 3** for this point. Given the intensity

of the research on photostimulation of neurons⁶⁰ and our prior studies in this area,⁶¹ this direction makes a lot of sense. With your permission, we shall explore the neuronal cell cultures as the next step in this work, for instance, to use circular polarization of incident light to decrease the excitation light intensity threshold of neurons.

Comment 3-24. *I note that the change in CD between unconstrained and constrained dimers appears to be more significant for NPs vs. NRs (I suspect that pre-existing multiple DNA linkages between NRs may be partly responsible for this). The manuscript would be strengthened if the authors addressed possible sources for the extent of each CD change in the supplementary information.*

Reply 3-24: This is accurate. We also think that dimers from NRs are ‘stiffer’ than those from NPs. We attribute the increased stiffness to stronger vdW interactions and thicker polymer coating, but some double bridges cannot be excluded.

Comment 3-25. *In Figure 5C and 5D, descriptions of the sample size should be reported, along with a description of what the error bars represent (i.e., standard error of the mean, standard deviation) in the figure legend. Units for the ROS intensity measured on the y-axis should also be included.*

Reply 3-25: The experiments for each group were run in triplicate; the error bars in the **Figure 5C,D** represent the standard deviation. We added the units for the singlet oxygen generation intensity measured on the y-axis in the **Figure 5C,D**.

Comment 3-26. *Were the numbers of DNA linkers and TAT peptides per particle empirically measured as part of this study, or were they assumed from previously published synthetic methods?*

Reply 3-26: We used the fluorescence method to measure the average number of DNA linkers and TAT peptides per particle according to references.^{30,31,62} The detailed procedure for determination of the stoichiometry of DNA/TAT constructs with NPs was added to *Methods*.

Comment 3-27. *Were NRs without PS-PAA shells tested? If so, what were the key findings for the unconstrained NR dimers?*

Reply 3-27. NRs without PA-PAA shells were also tested. These NR assemblies were incubated with HeLa cells for different times after being modified with PEG layer and TAT cell-penetrating peptides. However, due to the mass being higher than that of NPs, the NRs are not stable in the cytosol. From the bio-TEM below, we can see that the NR dimers aggregate in cells and thus cannot reconfigure.

Figure R16 (Figure S23) The bio-TEM image of NR dimer without PS-PAA coating after incubated with HeLa cells for 8 h.

Minor Comments for Supplementary Information

Comment 3-28. *Line 18: While it is well-known, the specific citrate reduction method should be cited.*

Reply 3-28: Thank you for reminding us about it. We agree and added the references into the *Supplementary Information* to give the original and slightly modified methods of this synthesis adapted to DNA-bridged superstructures.

1. Frens, G. Controlled Nucleation for the Regulation of the Particle Size in Monodisperse Gold Suspensions. *Nat. Phys. Sci.* **241**, 20–22 (1973).

2. Xu, L. *et al.* New Synthesis Strategy for DNA Functional Gold Nanoparticles. *J. Phys. Chem. C* **115**, 3243–3249 (2011).

Comment 3-29. *Line 22: For the purposes of replication, the centrifugation conditions (rcf, time, etc.) should be included.*

Reply 3-29: We have added the detailed centrifugation conditions into the *Supplementary Information*.

Comment 3-30: *Line 127: the word “can” should be deleted.*

Reply 3-30: Done.

Comments 3-31. *Line 179: “z-potential” should be changed to “ ζ -potential” since “z” is representative of another variable in the equations.*

Reply 3-31: Very true. We changed the “z-potential” to “ ζ -potential” .

Comment 3-31. *Regarding Figure S4: the change in intracellular NP distribution is quite abrupt between 18 and 24 hours. Can the authors explain why agglomeration appears to happen so suddenly? The same phenomenon is observe in Figures S21 and S22.*

Reply 3-31: Abrupt change in the agglomeration state of NP dimers is possible. It can be associated with, for instance, relatively slow cellular response resulting in the accumulation of the endonucleases.

Comment 3-32. *Regarding Figure S5: panel C shows an increase of ~ 0.1 OD over the first two hours once excess NP dimers are washed out. I am curious as to why an increase of a similar magnitude is not observed in panel B. Granted that the baseline absorbance is higher when excess dimers are not washed out, shouldn't there be an increase of the amount of NP dimers within the cells during the first two hours?*

Reply: The UV-Vis absorbance spectra in panel B are characteristic of intracellular NP dimers at each time point. In panel B, the total amount of intracellular and extracellular NP dimers is constant. As the interparticle distance is above 5 nm, its role in alteration of the intensity of plasmonic band become smaller. The amount of NP dimers acts as the dominant role in the intensity of UV-Vis spectra.

Comment 3-34. *Regarding Figure S12: it may be noted elsewhere, but the sample size should be reported for the measurement of dihedral angle distribution within the figure legend.*

Reply 3-34: We added the sample size to characterize the dihedral angle distribution in **Figure S16** as follows: “ *The statistical analysis of dihedral angles θ was obtained on the basis of 78, 76, 81, and 71 dimers from cryo-TEM tomography for the as-said four group, respectively*”.

Comment 3-35. *Line 33: a comma is needed to separate references 8 and 9.*

Reply: We have added a comma to separate references 8 and 9.

Comment 3-35 *Line 34: “localization of particles with accuracies comparable to cell sizes” is slightly unclear. Perhaps “spatial accuracies” would be more clear. Alternatively, “spectroscopic localization of particles with subcellular accuracy” may be clearer wording.*

Reply 3-35: We have revised the statements in the main text according to your suggestions.

Comment 3-37. *Line 63: a comma is needed to separate references 17 and 18.*

Reply: We have added the comma into the main text.

Comment 3-38. *The author names listed in reference 10 are not formatted consistently with the other references (i.e., last name, first initials)*

Reply 3-38: The format of reference 10 was revised as follows :

10. Sperling, R.A., Gil, P. R., Zhang, F. Zanella, M. and Parak, W. J. Biological applications of gold nanoparticles. *Chem. Soc. Rev.* **37**, 1896–1908 (2008).

Comment 3-39. *Line 157: It is difficult to tell since the text is justified, but there may*

be an additional space between the words “connecting” and “two.”

Reply 3-39: We have deleted the additional space between the words “connecting” and “two”.

Comment 3-40: *Line 171: protoporphyrin IX (PpIX) should be spelled out in addition to the abbreviation at its first mention within the text.*

Reply 3-40: We have added the full name of PpIX at its first mention in the main text.

Comment 3-41: *An explanation of the FITC/Texas Red imaging for the live/dead cell assays should be included in the Figure 5 legend to aid the rapid interpretation of the results.*

Reply 3-41: According to the suggestion by Reviewer, we have added the explanation in the legend for **Figure 5**.

Comment 3-42: *Recommended Additional References for the Manuscript*

1. S. Patskovsky, E. Bergeron, & M. Meunier, “Hyperspectral darkfield microscopy of PEGylated gold nanoparticles targeting CD44-expressing cancer cells,” J. Biophotonics 8(1-2), p.162-7 (2013).

2. E.D. SoRelle, O. Liba, J.L. Campbell, R. Dalal, C. Zavaleta, & A. de la Zerda, “A hyperspectral method to assay the microphysiological fates of nanomaterials in histological samples,” eLife 2016;10.7554/eLife.16352 (2016)

3. G.A. Roth, S. Tahiliani, N.M. Neu-Baker, S.A. Brenner, “Hyperspectral microscopy

as an analytical tool for nanomaterials,” WIRE Nanomed. & Nanobiotechnol. 7(4), p.565-79 (2015).

4. *E. Dulkeith, A.C. Morteani, T. Niedereichholz, T.A. Klar, J. Feldmann, S.A. Levi, F.C.J.M. van Veggel, D.N. Reinhoudt, M. Moller, & D.I. Gittins, “Fluorescence quenching of dye molecules near gold nanoparticles: radiative and non-radiative effects,” Phys. Rev. Lett. 89, 203002 (2002)*

5. *G. Schneider & G. Decher, “Distance-dependent fluorescent quenching on gold nanoparticles ensheathed with layer-by-layer assembled polyelectrolytes,” Nano Lett. 6(3), p.530-6 (2006)*

6. *J.R. Lakowicz, “Radiative decay engineering 5: metal-enhanced fluorescence and plasmon emission,” Anal. Biochem. 337, p.171-94 (2005).*

7. *K. Aslan, M. Wu, J.R. Lakowicz, & C.D. Geddes, “Fluorescent core-shell Ag@SiO₂ nanocomposites for metal-enhanced and single nanoparticle sensing platforms,” J. Am. Chem. Soc. 129(6), p.1524-5 (2007).*

8. *A.M. Funston, C. Novo, T.J. Davis, & P. Mulvaney, “Plasmon coupling of gold nanorods at short distances and in different geometries,” Nano Lett. 9(4), p. 1651-8 (2009).*

Reply 3-42: Thank you very much for giving additional references. We have added the related references into the manuscript.

References

- (1) Gore, J.; Bryant, Z.; Nöllmann, M.; Le, M. U.; Cozzarelli, N. R.; Bustamante, C. DNA Overwinds When Stretched. *Nature* **2006**, *442*, 836–839.
- (2) Ohta, S.; Glancy, D.; Chan, W. C. W. DNA-Controlled Dynamic Colloidal Nanoparticle Systems for Mediating Cellular Interaction. *Science (80-.)*. **2016**, *351*, 841–845.
- (3) Park, S. Y.; Lytton-Jean, A. K. R.; Lee, B.; Weigand, S.; Schatz, G. C.; Mirkin, C. A. DNA-Programmable Nanoparticle Crystallization. *Nature* **2008**, *451*, 553–556.
- (4) Zhang, L.; Lei, D.; Smith, J. M.; Zhang, M.; Tong, H.; Zhang, X.; Lu, Z.; Liu, J.; Alivisatos, A. P.; Ren, G. Three-Dimensional Structural Dynamics and Fluctuations of DNA-Nanogold Conjugates by Individual-Particle Electron Tomography. *Nat. Commun.* **2016**, *7*, 11083.
- (5) Lee, O. S.; Cho, V. Y.; Schatz, G. C. A- to B-Form Transition in DNA between Gold Surfaces. *J. Phys. Chem. B* **2012**, *116*, 7000–7005.
- (6) Zhang, H.; Govorov, a. O. Giant Circular Dichroism of a Molecule in a Region of Strong Plasmon Resonances between Two Neighboring Gold Nanocrystals. *Phys. Rev. B* **2013**, *87*, 1–8.
- (7) Shemer, G.; Krichevski, O.; Markovich, G.; Molotsky, T.; Lubitz, I.; Kotlyar, A. B. Chirality of Silver Nanoparticles Synthesized on DNA. *J. Am. Chem. Soc.* **2006**, *128*, 11006–11007.
- (8) Camden, J. P.; Dieringer, J. A.; Wang, Y.; Masiello, D. J.; Marks, L. D.; Schatz, G. C.; Van Duyne, R. P. Probing the Structure of Single-Molecule Surface-Enhanced Raman Scattering Hot Spots. *J. Am. Chem. Soc.* **2008**, *130*, 12616–12617.
- (9) Fang, Y.; Seong, N.-H.; Dlott, D. D. Measurement of the Distribution of Site Enhancements in Surface-Enhanced Raman Scattering. *Science (80-.)*. **2008**, *321*.
- (10) Ma, W.; Kuang, H.; Wang, L.; Xu, L.; Chang, W.-S.; Zhang, H.; Sun, M.; Zhu, Y.; Zhao, Y.; Liu, L.; *et al.* Chiral Plasmonics of Self-Assembled Nanorod Dimers. *Sci. Rep.* **2013**, *3*, 1934.
- (11) Amoozgar, Z.; Yeo, Y. Recent Advances in Stealth Coating of Nanoparticle Drug Delivery Systems. *Wiley Interdiscip. Rev. Nanomed. Nanobiotechnol.* **2012**, *4*, 219–233.
- (12) Jokerst, J. V.; Lobovkina, T.; Zare, R. N.; Gambhir, S. S. Nanoparticle PEGylation for Imaging and Therapy. *Nanomedicine (Lond)*. **2011**, *6*, 715–728.
- (13) Samoc, M.; Samoc, A.; Grote, J. G. Complex Nonlinear Refractive Index of DNA. *Chem. Phys. Lett.* **2006**, *431*, 132–134.
- (14) Cohen, G.; Eisenberg, H. Deoxyribonucleate Solutions: Sedimentation in a

- Density Gradient, Partial Specific Volumes, Density and Refractive Index Increments, and Preferential Interactions. *Biopolymers* **1968**, *6*, 1077–1100.
- (15) Banihashemian, S.; Periasamy, V.; Mohammadi, S.; Ritikos, R.; Rahman, S. Optical Characterization of Oligonucleotide DNA Influenced by Magnetic Fields. *Molecules* **2013**, *18*, 11797–11808.
- (16) Lindgren, M.; Hällbrink, M.; Prochiantz, A.; Langel, Ü. Cell-Penetrating Peptides. *Trends Pharmacol. Sci.* **2000**, *21*, 99–103.
- (17) Richard, J. P.; Melikov, K.; Vives, E.; Ramos, C.; Verbeure, B.; Gait, M. J.; Chernomordik, L. V.; Lebleu, B. Cell-Penetrating Peptides. A Reevaluation of the Mechanism of Cellular Uptake. *J. Biol. Chem.* **2003**, *278*, 585–590.
- (18) Guerrero-Martínez, A.; Auguié, B.; Alonso-Gómez, J. L.; Džolić, Z.; Gómez-Graña, S.; Žinić, M.; Cid, M. M.; Liz-Marzán, L. M. Intense Optical Activity from Three-Dimensional Chiral Ordering of Plasmonic Nanoantennas. *Angew. Chem. Int. Ed. Engl.* **2011**, *50*, 5499–5503.
- (19) Kuzyk, A.; Schreiber, R.; Zhang, H.; Govorov, A. O.; Liedl, T.; Liu, N. Reconfigurable 3D Plasmonic Metamolecules. *Nat. Mater.* **2014**, *13*, 862–866.
- (20) Ma, W.; Kuang, H.; Xu, L.; Ding, L.; Xu, C.; Wang, L.; Kotov, N. A. Attomolar DNA Detection with Chiral Nanorod Assemblies. *Nat. Commun.* **2013**, *4*, 2689.
- (21) Pelaz, B.; Jaber, S.; de Aberasturi, D. J.; Wulf, V.; Aida, T.; de la Fuente, J. M.; Feldmann, J.; Gaub, H. E.; Josephson, L.; Kagan, C. R.; *et al.* The State of Nanoparticle-Based Nanoscience and Biotechnology: Progress, Promises, and Challenges. *ACS Nano* **2012**, *6*, 8468–8483.
- (22) Dykman, L.; Khlebtsov, N.; Ochsner, E. H.; Rogoff, E. E.; Romano, R.; Hahn, E. W.; Liu, S.; Leech, D.; Ju, H.; Andreu, E. J.; *et al.* Gold Nanoparticles in Biomedical Applications: Recent Advances and Perspectives. *Chem. Soc. Rev.* **2012**, *41*, 2256–2282.
- (23) Cabuzu, D.; Cirja, A.; Puiu, R.; Grumezescu, A. Biomedical Applications of Gold Nanoparticles. *Curr. Top. Med. Chem.* **2015**, *15*, 1605–1613.
- (24) Lundqvist, M.; Stigler, J.; Elia, G.; Lynch, I.; Cedervall, T.; Dawson, K. A. Nanoparticle Size and Surface Properties Determine the Protein Corona with Possible Implications for Biological Impacts. *Proc. Natl. Acad. Sci. U. S. A.* **2008**, *105*, 14265–14270.
- (25) Monopoli, M. P.; Walczyk, D.; Campbell, A.; Elia, G.; Lynch, I.; Baldelli Bombelli, F.; Dawson, K. A. Physical–Chemical Aspects of Protein Corona: Relevance to *in Vitro* and *in Vivo* Biological Impacts of Nanoparticles. *J. Am. Chem. Soc.* **2011**, *133*, 2525–2534.
- (26) Safavi-Sohi, R.; Maghari, S.; Raoufi, M.; Jalali, S. A.; Hajipour, M. J.; Ghassempour, A.; Mahmoudi, M. Bypassing Protein Corona Issue on Active Targeting: Zwitterionic Coatings Dictate Specific Interactions of Targeting Moieties and Cell Receptors. *ACS Appl. Mater. Interfaces* **2016**, *8*, 22808–22818.

- (27) Lee, Y. K.; Choi, E.-J.; Webster, T. J.; Kim, S.-H.; Khang, D. Effect of the Protein Corona on Nanoparticles for Modulating Cytotoxicity and Immunotoxicity. *Int. J. Nanomedicine* **2015**, *10*, 97–113.
- (28) Schöttler, S.; Becker, G.; Winzen, S.; Steinbach, T.; Mohr, K.; Landfester, K.; Mailänder, V.; Wurm, F. R. Protein Adsorption Is Required for Stealth Effect of Poly(ethylene Glycol)- and Poly(phosphoester)-Coated Nanocarriers. *Nat. Nanotechnol.* **2016**, *11*, 372–377.
- (29) Pino, P. del; Pelaz, B.; Zhang, Q.; Maffre, P.; Nienhaus, G. U.; Parak, W. J. Protein Corona Formation around Nanoparticles – from the Past to the Future. *Mater. Horiz.* **2014**, *1*, 301–313.
- (30) Lewin, M.; Carlesso, N.; Tung, C. H.; Tang, X. W.; Cory, D.; Scadden, D. T.; Weissleder, R. Tat Peptide-Derivatized Magnetic Nanoparticles Allow in Vivo Tracking and Recovery of Progenitor Cells. *Nat. Biotechnol.* **2000**, *18*, 410–414.
- (31) Kumar, S.; Aaron, J.; Sokolov, K. Directional Conjugation of Antibodies to Nanoparticles for Synthesis of Multiplexed Optical Contrast Agents with Both Delivery and Targeting Moieties. *Nat. Protoc.* **2008**, *3*, 314–320.
- (32) Krpetić, Ž.; Saleemi, S.; Prior, I. a.; Sée, V.; Qureshi, R.; Brust, M. Negotiation of Intracellular Membrane Barriers by TAT-Modified Gold Nanoparticles. *ACS Nano* **2011**, *5*, 5195–5201.
- (33) Sadelain, M.; Papapetrou, E. P.; Bushman, F. D. Safe Harbours for the Integration of New DNA in the Human Genome. *Nat Rev Cancer* **2012**, *12*, 51–58.
- (34) Elstner, M.; Hobza, P.; Frauenheim, T.; Suhai, S.; Kaxiras, E. Hydrogen Bonding and Stacking Interactions of Nucleic Acid Base Pairs: A Density-Functional-Theory Based Treatment. *J. Chem. Phys.* **2001**, *114*, 5149–5155.
- (35) Wang, S.; Huang, P.; Nie, L.; Xing, R.; Liu, D.; Wang, Z.; Lin, J.; Chen, S.; Niu, G.; Lu, G.; *et al.* Single Continuous Wave Laser Induced Photodynamic/plasmonic Photothermal Therapy Using Photosensitizer-Functionalized Gold Nanostars. *Adv. Mater.* **2013**, *25*, 3055–3061.
- (36) Idris, N. M.; Gnanasammandhan, M. K.; Zhang, J.; Ho, P. C.; Mahendran, R.; Zhang, Y. In Vivo Photodynamic Therapy Using Upconversion Nanoparticles as Remote-Controlled Nanotransducers. *Nat. Med.* **2012**, *18*, 1580–1585.
- (37) Murphy, C. J.; Thompson, L. B.; Alkilany, A. M.; Sisco, P. N.; Boulos, S. P.; Sivapalan, S. T.; Yang, J. A.; Chernak, D. J.; Huang, J. The Many Faces of Gold Nanorods. *J. Phys. Chem. Lett.* **2010**, *1*, 2867–2875.
- (38) Hirsch, L. R.; Stafford, R. J.; Bankson, J. A.; Sershen, S. R.; Rivera, B.; Price, R. E.; Hazle, J. D.; Halas, N. J.; West, J. L. Nanoshell-Mediated near-Infrared Thermal Therapy of Tumors under Magnetic Resonance Guidance. *Proc. Natl. Acad. Sci. U. S. A.* **2003**, *100*, 13549–13554.
- (39) Bayraktar, H.; Ghosh, P. S.; Rotello, V. M.; Knapp, M. J. Disruption of Protein-

- Protein Interactions Using Nanoparticles: Inhibition of Cytochrome c Peroxidase. *Chem. Commun. (Camb)*. **2006**, 1390–1392.
- (40) Cifuentes-Rius, A.; Puig, H. de; Kah, J. C. Y.; Borros, S.; Hamad-Schifferli, K. Optimizing the Properties of the Protein Corona Surrounding Nanoparticles for Tuning Payload Release. **2013**.
- (41) Kim, C. S.; Li, X.; Jiang, Y.; Yan, B.; Tonga, G. Y.; Ray, M.; Solfiell, D. J.; Rotello, V. M. Cellular Imaging of Endosome Entrapped Small Gold Nanoparticles. *MethodsX* **2015**, *2*, 306–315.
- (42) Huang, X.; El-Sayed, I. H.; Qian, W.; El-Sayed, M. a. Cancer Cell Imaging and Photothermal Therapy in the near-Infrared Region by Using Gold Nanorods. *J. Am. Chem. Soc.* **2006**, *128*, 2115–2120.
- (43) SoRelle, E. D.; Liba, O.; Campbell, J. L.; Dalal, R.; Zavaleta, C. L.; de la Zerda, A. A Hyperspectral Method to Assay the Microphysiological Fates of Nanomaterials in Histological Samples. *Elife* **2016**, *5*, e16352.
- (44) Patskovsky, S.; Bergeron, E.; Meunier, M. Hyperspectral Darkfield Microscopy of PEGylated Gold Nanoparticles Targeting CD44-Expressing Cancer Cells. *J. Biophotonics* **2015**, *8*, 162–167.
- (45) Roth, G. a.; Tahiliani, S.; Neu-Baker, N. M.; Brenner, S. a. Hyperspectral Microscopy as an Analytical Tool for Nanomaterials. *Wiley Interdiscip. Rev. Nanomedicine Nanobiotechnology* **2015**, *7*, 565–579.
- (46) Ding, Y.; Jiang, Z.; Saha, K.; Kim, C. S.; Kim, S. T.; Landis, R. F.; Rotello, V. M. Gold Nanoparticles for Nucleic Acid Delivery. *Mol. Ther.* **2014**, *22*, 1075–1083.
- (47) Rosi, N. L.; Giljohann, D. a; Thaxton, C. S.; Lytton-Jean, A. K. R.; Han, M. S.; Mirkin, C. a. Oligonucleotide-Modified Gold Nanoparticles for Intracellular Gene Regulation. *Science* **2006**, *312*, 1027–1030.
- (48) Louit, G.; Asahi, T.; Tanaka, G.; Uwada, T.; Masuhara, H. Spectral and 3-Dimensional Tracking of Single Gold Nanoparticles in Living Cells Studied by Rayleigh Light Scattering Microscopy. *J. Phys. Chem. C* **2009**, *113*, 11766–11772.
- (49) Ba, H.; Rodríguez-Fernández, J.; Stefani, F. D.; Feldmann, J. Immobilization of Gold Nanoparticles on Living Cell Membranes upon Controlled Lipid Binding. *Nano Lett.* **2010**, *10*, 3006–3012.
- (50) Lim, H.-T.; Murukeshan, V. M. Spatial-Scanning Hyperspectral Imaging Probe for Bio-Imaging Applications. *Rev. Sci. Instrum.* **2016**, *87*, 33707.
- (51) Haas, B. L.; Matson, J. S.; DiRita, V. J.; Biteen, J. S. Imaging Live Cells at the Nanometer-Scale with Single-Molecule Microscopy: Obstacles and Achievements in Experiment Optimization for Microbiology. *Molecules* **2014**, *19*, 12116–12149.
- (52) Patskovsky, S.; Bergeron, E.; Rioux, D.; Meunier, M. Wide-Field Hyperspectral

- 3D Imaging of Functionalized Gold Nanoparticles Targeting Cancer Cells by Reflected Light Microscopy. *J. Biophotonics* **2015**, *8*, 401–407.
- (53) Lee, J.; Govorov, A. O.; Dulka, J.; Kotov, N. a. Bioconjugates of CdTe Nanowires and Au Nanoparticles: Plasmon-Exciton Interactions, Luminescence Enhancement, and Collective Effects. *Nano Lett.* **2004**, *4*, 2323–2330.
- (54) Lee, J.; Hernandez, P.; Lee, J.; Govorov, A. O.; Kotov, N. a. Exciton–plasmon Interactions in Molecular Spring Assemblies of Nanowires and Wavelength-Based Protein Detection. *Nat. Mater.* **2007**, *6*, 291–295.
- (55) Lee, J.; Govorov, A. O.; Kotov, N. a. Bioconjugated Superstructures of CdTe Nanowires and Nanoparticles: Multistep Cascade Förster Resonance Energy Transfer and Energy Channeling. *Nano Lett.* **2005**, *5*, 2063–2069.
- (56) Zhao, Y.; Xu, L.; Ma, W.; Wang, L.; Kuang, H.; Xu, C.; Kotov, N. A. Shell-Engineered Chiroplasmonic Assemblies of Nanoparticles for Zeptomolar DNA Detection. *Nano Lett.* **2014**, *14*, 3908–3913.
- (57) Rodríguez-Lorenzo, L.; de la Rica, R.; Álvarez-Puebla, R. A.; Liz-Marzán, L. M.; Stevens, M. M. Plasmonic Nanosensors with Inverse Sensitivity by Means of Enzyme-Guided Crystal Growth. *Nat. Mater.* **2012**, *11*, 604–607.
- (58) Funston, A. M.; Novo, C.; Davis, T. J.; Mulvaney, P. Plasmon Coupling of Gold Nanorods at Short Distances and in Different Geometries. *Nano Lett.* **2009**, *9*, 1651–1658.
- (59) Liz-marzan, L. M.; Giersig, M.; Mulvaney, P. Synthesis of Nanosized Gold - Silica Core - Shell Particles. *Langmuir* **1996**, *7463*, 4329–4335.
- (60) Wu, X.; Zhang, Y.; Takle, K.; Bilsel, O.; Li, Z.; Lee, H.; Zhang, Z.; Li, D.; Fan, W.; Duan, C.; *et al.* Dye-Sensitized Core/Active Shell Upconversion Nanoparticles for Optogenetics and Bioimaging Applications. *ACS Nano* **2016**, *10*, 1060–1066.
- (61) Pappas, T. C.; Wickramanyake, W. M. S.; Jan, E.; Motamedi, M.; Brodwick, M.; Kotov, N. A. Nanoscale Engineering of a Cellular Interface with Semiconductor Nanoparticle Films for Photoelectric Stimulation of Neurons. *Nano Lett.* **2007**, *7*, 513–519.
- (62) Demers, L. M.; Mirkin, C. a; Mucic, R. C.; Reynolds, R. a; Letsinger, R. L.; Elghanian, R.; Viswanadham, G. Based Method for Determining the Surface Coverage and Hybridization Efficiency of Thiol - Capped Oligonucleotides Bound to Gold Thin Films and Nanoparticles. *Anal. Chem.* **2000**, *72*, 5535–5541.
- (63) Gérard, V. A.; Gun'ko, Y. K.; Defrancq, E.; Govorov, A. O.; Storhoff, J. J.; Elghanian, R.; Mirkin, C. A.; Mucic, R. C.; Letsinger, R. L.; Hamad-Schifferli, K.; *et al.* Plasmon-Induced CD Response of Oligonucleotide-Conjugated Metal Nanoparticles. *Chem. Commun.* **2011**, *47*, 7383.

Reviewers' comments:

Reviewer #1 (Remarks to the Author):

I have read the responses of the authors to my comments and the other two reviewers.

I would like to try to be brief this time and write the main criticism that I still have to three central points:

1. The mechanism leading to the appearance of the CD in the DNA linked NP dimers. The authors have estimated the magnitude of the hot-spot effect in the dimers, and estimate it to be too small to explain the observed CD signal. Let's assume that this is true.

Still, I have a serious problem with the proposed twisting model. I see that I neglected to explain my main objection to this model proposed by the authors: According to the illustrations and electron tomography results the dimers always form nearly parallel configurations with both their long axes arranged close to parallel. And according to this the two DNA-thiol anchors should bind close to the center ("equatorial") regions in the two NPs. But this event should be statistically quite rare. Why should the thiols not bind to random locations at each NP? This should also be the case for the nanorods. The low magnification TEM images of the NPs such as Fig. 1B, Fig. 2A do not enable estimation of the dimer configurations. However, for the nanorods, it is easier to see, for example in Figs. S24, S28 that many of them form different conformations than (nearly?) parallel ones. And in the nanorods there is a stronger preference towards parallel conformations due to much higher VdW energy gain in parallel configurations relative to NP dimers.

So, is the selection of dimers probed by electron tomography biased towards specific configurations? Or is there a special driving mechanism that would cause the thiolated DNA linkers to specifically bind around the equatorial area in the NPs? It was actually seen in gold NRs that there is preference of thiol binding to NR edges, probably due to the smaller radius of curvature there, which leaves more free volume in the coating molecules to allow easier approach of the thiolated molecules to the NR surface. I did a survey of the aspect ratio of the NPs shown in the tomography images in the paper and SI. Their aspect ratios were in the range of 1.5-2.0 and even higher, while in the paper the average aspect ratio is reported to be tightly distributed around 1.3. This is an indication that the tomography images were biased towards particular dimers.

This is why I find it difficult to believe the proposed twisting mechanism. The other authors also raised concerns, such as multiple DNA binding in the case of the NRs. Especially for conditions that favor dimers over other particle oligomers and with average of 1.6 DNA molecules per dimer. Hence, I still see a hot-spot related mechanism (or any other, not twist related mechanism) as more probable than the proposed one.

2. Signal reversal on dimer penetration into the cells. I originally missed the fact that the cells are immobilized and consist of a highly viscous medium. As reviewer 2 pointed out for solid-like media, linear dichroism effects become important and may dominate the measured CD spectrum. The authors indeed show in their response 2-1, that they have measured LD signal at the plasmon resonance wavelength of the NPs in the cells. This could pose a problem for the CD measurement, since even relatively low LD might form a strong interference to CD measurements. I am not sure if this would be the reason for CD inversion, but as stated above also the twisting model is not convincing at this point.

3. The large differences observed between the two circular polarizations in killing cells with photodynamic therapy. The authors did not really answer my comment 1-5. Indeed, cell killing is not a direct result of the dimer absorption (hence also CD) but the result of production of ROS due to absorption of PpIX. Hence, the difference in the cell killing should come from difference in absorption of the PpIX of the two circular polarizations, which in turn, would lead to a difference in ROS production, right? In that case the authors probably rely on the induction of CD from the plasmonic system to the molecular PpIX system, right? These effects are also typically fairly small and would probably not go over 1% difference. So, still we are left with the mystery of where do effects of 200-300% difference come from? The authors should suggest an explanation for that. Why not show the CD spectra of the PpIX interacting with the dimers? -this should provide a direct estimate of the difference in ROS production rates.

In their response they wrote that it is because the apoptosis is the result of forming ROS INSIDE THE CELL. I do not view this as any sort of explanation.

Those are my main concerns. In short, again: 1. There is no reason to believe that the dimers form a uniform set of twisted pairs when there is no reason to believe that the thiolated DNA preferentially binds at the equatorial zone of the NPs. 2. Then the inversion of the CD signal remains also unexplained. 3. The 200-300% difference in cell killing cannot be simply explained by the optical activity of the dimers and its influence on the photodynamic therapy reagent, which would have a max. 1% effect.

I do not see how the paper can be published before really answering these concerns.

Reviewer #2 (Remarks to the Author):

After a careful reading of the new version of the manuscript, I find that the authors have conveniently explained all the questions and comments that were left hanging by the referees. The proposed mechanism of nanoparticle uptake is fully convincing, based on the optical activity experiments, the electron microscopy analysis, and the new version of the theoretical model. The concept of using circular dichroism as a novel tool to understand the interaction between plasmonic nanoparticles and the intracellular organelles is simply exciting. In my opinion, Nature Communications seems a perfect medium for publication of this conceptually novel investigation, which will be appealing to a broad audience within general and applied science.

Reviewer #3 (Remarks to the Author):

The authors took great care in addressing the points raised in my original review. I appreciate the additional experiments they have provided, the references they have added, and the clarifications they have made to the article text. Because the work is clear, comprehensive, and interesting to a wide audience, I recommend that the article "Intracellular Localization of Nanoparticle Dimers by Chirality Reversal" be published in Nature Communications.

Responses to Reviewer 1

Reviewer 1 (verbatim).

I have read the responses of the authors to my comments and the other two reviewers. I would like to try to be brief this time and write the main criticism that I still have to three central points:

Comment 1-1: The mechanism leading to the appearance of the CD in the DNA linked NP dimers. The authors have estimated the magnitude of the hot-spot effect in the dimers, and estimate it to be too small to explain the observed CD signal. Let's assume that this is true.

Still, I have a serious problem with the proposed twisting model. I see that I neglected to explain my main objection to this model proposed by the authors: According to the illustrations and electron tomography results the dimers always form nearly parallel configurations with both their long axes arranged close to parallel. And according to this the two DNA-thiol anchors should bind close to the center ("equatorial") regions in the two NPs. But this event should be statistically quite rare. Why should the thiols not bind to random locations at each NP? This should also be the case for the nanorods. The low magnification TEM images of the NPs such as Fig. 1B, Fig. 2A do not enable estimation of the dimer configurations. However, for the nanorods, it is easier to see, for example in Figs. S24, S28 that many of them form different conformations than (nearly?) parallel ones. And in the nanorods there is a stronger preference towards parallel conformations due to much higher VdW energy gain in parallel configurations relative to NP dimers.

So, is the selection of dimers probed by electron tomography biased towards specific configurations? Or is there a special driving mechanism that would cause the thiolated DNA linkers to specifically bind around the equatorial area in the NPs? It was actually seen in gold NRs that there is preference of thiol binding to NR edges, probably due to the smaller radius of curvature there, which leaves more free volume in the coating molecules to allow easier approach of the thiolated molecules to the NR surface.

Reply 1-1. For clarity, **Figures 1B, 2A, S24, and S28** referred to by **Reviewer 1**, are reproduced below (**Figure R1**). It appears that the first point raised by the **Reviewer** is that in these images the long axes nanoparticles and nanorods are oriented in nearly parallel configurations and not in the twisted configuration.

Not knowing the background of the Reviewer, we need to point out – perhaps unnecessarily – that the images in **Figures 1B (inset), S24, and S28** are not electron tomography data but are bright field transmission electron microscopy. They represent a thin, in-focus slice of the 3D object. They, therefore, cannot be and were not used to evaluate the dihedral angle and therefore validate/invalidate the dimer twist

model.

Figure S28 indeed shows the images of the dimers inside the cells and in a variety of assembly states of nanorods. Importantly, **Figures S28 at 2 h, 8 h, 12 h, and 18 h** show that the assemblies retain the original stoichiometry and conformation of twisted dimer of dimers for the long period of time inside the cells. They do not contradict in any way the model of twisted dimers. **Figures S28 at 24 h and 48 h** are the ones that **Reviewer 1** might be referring to when stating “..that many of them form different conformations than (nearly?) parallel ones. These images show the state of the dimers *digested by HeLa cells*, agglomerated in exosomes. This state occurs much later than 2 h (**Figures 1C, 1D**) when switching of the dimer configuration occurs. These agglomerates are formed when a lot of the DNA bridges are cut by endonucleases inside the cells.

Please note that true 3D images of the dimers taken by cryo electron tomography when the assembled nanoparticle pairs are in their native environment presented in **Figures 2C, S14, and S15**, show with great clarity that the long axes are not parallel and the dimers have distinct twist. The angle of the twist changes with the surrounding environment. **Figures 2D, S23, and S16** make the same point from a perspective of statistical analysis.

It appears that the second point raised by **Reviewer 1** is about equatorial binding of thiols. The **Reviewer** argues that “... this event should be statistically quite rare. Why should the thiols not bind to random locations at each NP?” We have to say that these arguments are quite confusing because further along in the list of critical comments **Reviewer 1** contradicts his/her assertion of random binding by arguing about the preferential binding in the ends of the nanorods. Anyway, we shall try to address it in a consistent manner.

First of all, the perfection of binding strictly in the middle of the ellipsoid is not required for the dihedral angle between them and polarization rotation to emerge. Following **Reviewer 1's** point about van der Waals attraction (“..there is a stronger preference

towards parallel conformations due to much higher VdW energy gain in parallel configurations relative to NP dimers’.”), the attractive forces between the particles will bring them together, even when the ligands are bound off-center. Besides that, the equatorial binding is *the most probable localization of the surface ligands* for geometrical reasons. The surface area and circumference are the greatest in the equatorial part of the ellipsoid (**Figure R2**). Hence, the probability binding of thiol and other ligands is more probable in this area than in poles of the ellipsee.

Similar effects of regioselectivity are known in other areas of chemistry. For instance, bis-adducts of fullerenes in *equatorial* configurations have the highest yield among other derivatives due to the identical geometrical reasoning and corresponding quantum mechanical effects based on electron delocalization along the longest circumference.¹

The model calculations using a line charge model (**Figure S31**, Revision 2; **Figure 30**, Revision 1) or MD simulations (**Figure S35**, Revision 2) are based on the binding in the equatorial section because we need to make some reasonable assumptions to perform the calculations.

The preferential end-bindings that **Reviewer 1** refers to occurs in CTAB-stabilized nanorods.² This work is focused on chiroptical phenomena with 20 nm Au *nanoparticles made with citrate*, not CTAB (**SI, MATERIALS AND METHODS**, first sentence). The NRs are used in this work only for benchmarking. If the end-binding would dominate for nanorods, one would see predominantly chains of nanorods, not “...nearly parallel configurations” of nanorods.

Comment 1-2: I did a survey of the aspect ratio of the NPs shown in the tomography images in the paper and SI. Their aspect ratios were in the range of 1.5-2.0 and even higher, while in the paper the average aspect ratio is reported to be tightly distributed around 1.3. This is an indication that the tomography images were biased towards particular dimers.

This is why I find it difficult to believe the proposed twisting mechanism. The other authors also raised concerns, such as multiple DNA binding in the case of the NRs. Especially for conditions that favor dimers over other particle oligomers and with average of 1.6 DNA molecules per dimer.

Hence, I still see a hot-spot related mechanism (or any other, not twist related mechanism) as more probable than the proposed one.

Reply 1-2: It appears that the third point raised by **Reviewer 1** is that the aspect ratios in the images that he/she surveyed were “...were in the range of 1.5-2.0 and even higher, while in the paper the average aspect ratio is reported to be tightly distributed around 1.3. This is an indication that the tomography images were biased towards particular dimers”. We respect and appreciate the right of the **Reviewer** to question our experimental methods and search for flaws. We are happy to address this criticism and concern about the statistical validity of the aspect ratio data reported in the manuscript. In response to this comment, we made a corresponding note to **Figures S14** and **S15**, and added to the legend the comment that *the appearance of higher-than-average anisotropy compared to bright-field TEM images is due to the ‘missing wedge’ problem associated with the limitation of the tilt angle to $\pm 60^\circ$ used in 3D image reconstruction.*

In this work, a tomographic tilt series of 104 projections with equal-slope increments and a tilt range of $\pm 60^\circ$ were acquired from nanoparticle dimers. Specimens cannot be tilted beyond $\pm 60^\circ$, preventing acquisition of data from that ‘missing wedge’ between 60° and 90° . Owing to the ‘missing wedge’ problem, reconstructed data can produce the larger anisotropy and decrease the resolution of 3D reconstruction images and cause the blurred edges.¹⁻⁵ Because of the above reasons, the aspect ratio of Au NPs from TEM tomography was calculated to be 1.5 ± 0.3 , which is larger than that based on the bright-field TEM images.

The statistical analysis of aspect ratio was obtained on the basis of bright-field TEM images. The long and short axes of 300 nanoparticles were measured and the data are presented in (**Figure 3A**).

Detailed consideration of the hot-spot mechanisms are given in **Figures S17** and **S18**, accompanied by extensive discussion in the Supplementary Information. As shown in **Figure S17**, the circular dichroism due to hot-spot generation contributes only a minor part to the total chiroptical activity in this system.

We also obtained independent validation of the twisted dimer model by Molecular Dynamics simulations. The data presented in **Figure 5** match the conclusion made from the line charge model (**Figure 4**).

Comment 1-3: Signal reversal on dimer penetration into the cells. I originally missed the fact that the cells are immobilized and consist of a highly viscous medium. As reviewer 2 pointed out for solid-like media, linear dichroism effects become important and may dominate the measure CD spectrum. The authors indeed show in their response 2-1, that they have measured LD signal at the plasmon resonance wavelength of the NPs in the cells. This could pose a problem for the CD measurement, since even relatively low LD might form a strong interference to CD measurements. I am not sure if this would be the reason for CD inversion, but as stated above also the twisting model is not convincing at this point.

Reply 1-3: The CD spectra of the cells in **Figure 1** were measured from the solution, not immobilized onto solid-like substrates. Therefore, the randomization of the NP dimers in 4π space is still present. Therefore, LD cannot be responsible for the switching of the sign of circular dichroism in dispersion. Comments about this point were added to **Figures 1** and **2**.

Nevertheless, we carried out an experiment of the linear dichroism of NP dispersions with cells, gradually increasing incubation time over a period of 12 h (**Figure R3**), and the results displayed nothing but noise.

Figure R3 | Temporal progressions of linear dichroism (LD) spectra for NP dimers being incubated with HeLa cells. **A-B**, LD spectra of individual NPs and NP dimers in the PBS buffer. **C-F**, LD spectra of NP dimers incubated with HeLa cells within 2 h after (**C, D**) and before (**E, F**) the removal of extracellular dimers. Note that **A, C, and E** were the original LD spectra of NP dimers, **B, D, and F** were smoothed from **A, C, and E** by the Savitzky-Golay method.

Comment 1-4: The large differences observed between the two circular polarizations in killing cells with photodynamic therapy. The authors did not really answer my comment 1-5. Indeed, cell killing is not a direct result of the dimer absorption (hence also CD) but the result of production of ROS due to absorption of PpIX. Hence, the difference in the cell killing should come from difference in absorption of the PpIX of the two circular polarizations, which in turn, would lead to a difference in ROS production, right? In that case the authors probably rely on the induction of CD from the plasmonic system to the molecular PpIX system, right? These effects are also

typically fairly small and would probably not go over 1% difference. So, still we are left with the mystery of where do effects of 200-300% difference come from? The authors should suggest an explanation for that. Why not show the CD spectra of the PpIX interacting with the dimers? -this should provide a direct estimate of the difference in ROS production rates. In their response they wrote that it is because the apoptosis is the result of forming ROS INSIDE THE CELL. I do not view this as any sort of explanation.

Reply 1-4: The dramatic increase of the cell death ratio for the LCP compared to RCP is indeed quite remarkable and hence, we thought that *Nature Communication* will be appropriate journal for publication of these results. We are happy to give a more detailed explanations and appreciate the logic expressed by **Reviewer 1** in the comment above. However, the **Reviewer** does not include two essential factors in the consideration that are essential for understanding such effect:

(a) ROS have very short limited lifetime

and

(b) metabolic activity and rupture of cells results in NP dimers present in the media.

The lifetime of singlet oxygen, *i.e.* the excited state of the molecule of O₂ that exists as a triplet state at the ground level, is 200 ns in biological media.⁶ During this lifetime singlet oxygen can diffuse over the distance of 10-50 nm depending on the biological tissue.^{7,8} Therefore, the vast majority of ROS generated outside the cells and are “wasted” and so are the photons that were used on their production. Exocytosis of the live cells (**Figure S6**) and rupture of the dead cells will always result in significant concentration of the dimers and other assemblies in the buffer. These are the particles that will absorb light but will not contribute to the apoptosis induction because ROS will be quickly destroyed by the media components. Note that the dimers dispersed in the buffer will predominantly adsorb RCP.

On the other hand, the molecules of singlet oxygen generated inside the cells are the ones that are the ones that destroy the cells. Consequently, the photons that are adsorbed inside the cells are the ones that we need to be interested. Therefore, the photons adsorbed inside the cells by the dimers with left helicity (**Figure 2C**), that is

LCP photons, have greater “potency” than the LCP photons that are predominantly adsorbed/scattered outside of the cells. The direct estimate of the production rate inside the cells will be an entirely new project with its own experimental challenges and relevant benchmarks that will require in our estimate at least 12 month. We believe it should be relegated to a separate manuscript.

Reviewers' comments:

Reviewer #1 (Remarks to the Author):

No further comments

[Editor's note: The following comments by Reviewer #3 were paraphrased from a conversation between the editor and this reviewer.]

Replies to Reviewer's Comments

Reviewer #3:

Comment 3-1: You may want to consider performing some kind of control photosensitizer experiments for dimers without cell penetrating peptides to quantitatively assess cell death as a function of intracellular vs extracellular localization.

Reply 3-1: We thank the reviewer for his/her encouraging comments and appreciate it. We have added excess control experiments as the request. First, the NP dimer **without TAT** cell penetrating peptides were prepared and incubated with HeLa cells for 18h. The cell with extracellular NP dimer were irradiated by CPL for 30min at 532nm, 5 mW/cm². As shown by the live/dead MTT assay the death ratio of no-TAT NP dimer for LCP is $10.78 \pm 1.1\%$ while RCP is $53.54 \pm 3.7\%$. The corresponding data for the dimer that carry TAT, the death ratio for LCP is $93.5 \pm 3.5\%$ while RCP is $31.64 \pm 2.3\%$ (**Figure R1A**). Confocal images of live/dead assay confirm the results (**Figure R2**). Only a small amount of dead cells could be observed in NP dimer conjugated with PpIX under LCP, while about half of the cells were killed under RCP irradiation. This result demonstrated that the death ratio of no-TAT NP dimer conjugated with PpIX was higher under RCP than LCP, which is opposite with the dimer that carry TAT and are transported into the cell (**Figure R1A** and **R3**).

These additional data concur with the phenomenological model advocated in the previous version of the manuscript. The NP dimer without cell penetrating peptides cannot be transported into the cell efficiently and therefore most of the dimers are

residing outside the cell. The no-TAT NP dimer display negative CD peak (**Figure R4**) which means that absorption of RCP photons is stronger than LCP ones. Correspondingly, the photodynamic cell killing rate with RCP photons by the dimers located outside the cells was higher than with LCP. Also note that the overall killing rate in this conditions is almost two times lower than for dimers bearing the TAT peptides and illuminated with LCP photons, which originates from the less efficient delivery of ROS to the cells due to the longer diffusion path and short lifetime of the singlet oxygen.

Concurrently, we also prepared and tested photodynamic therapy with **PS-PAA-constrained** NR dimer to provide additional validation and benchmark to our understanding of the photo-stimulated phenomena taking place in this system. HeLa cells after being incubated for 18h with fixed dimers were irradiated CPL for 30min at 660nm, 5 mW/cm². The fixed dimers are supposed to have stronger RCP adsorption than LCP regardless whether they are inside or outside of the cell. Indeed, the dimers with and without TAT peptide showed higher death rate for RCP than for LCP (**Figure R1B**): with TAT peptides LCP - 47.34±1.8%, RCP- 94.5±2.5% while without TAT peptides LCP - 11.26±0.6%, RCP - 58.47±2.1%. Notice again, the death ratio of NR dimers with TAT is higher than for those without TAT for both polarizations of light because the generation of ROS inside the cells is more efficient in disrupting their functions than for ROS outside the cells.

Confocal images of the cell cultures after the photodynamic therapy with fixed

dimers match the killing rates enumerated by MTT assays. The fixed NR dimer without TAT cell penetrating peptides showed that the RCP photons generate higher percentage of apoptotic cells than LCP (Figure R5). The same is true in this case for fixed dimers with TAT that penetrate inside the cell (Figure R6). The confocal images also visually confirm that the death ratio (density of red cells) for fixed NR dimer without TAT is lower than the dimers with them due to the limited life-time of ROS.

Corresponding changes were made in the main text and Supporting information and highlighted in red.

Figure R1 (Figure 7 C-D). The death ratio of HeLa cells incubated with (A) NP dimers, (B) NR dimers under different polarized light irradiation determined by MTT assay. The NP or NR dimers with / without cell penetrating peptides modified on the surface were denoted by with TAT / without TAT.

Figure R2 (Figure 7 A). The confocal microscopy images for the live (green) /dead (red) assays of HeLa cells after incubation with NP dimers conjugated with PpIX (without cell penetrating peptides labeled on the NP surface and denoted by **without TAT**) and irradiation under (A) LCP and (B) RCP for with 532 nm photons for 30 min (5 mW/cm^2). Scale bar 200 μm .

Figure R3 (Figure 6A). The confocal microscopy images for the live (green) /dead (red) assays of HeLa cells after incubation with NP dimers conjugated with PpIX (with cell penetrating peptides labeled on the NP surface and denoted by **with TAT**) and irradiation under (A) LCP and (B) RCP for with 532 nm photons for 30 min (5 mW/cm^2). Scale bar 200 μm .

Figure R4 (Figure 1B). CD spectra of individual NPs, HeLa cells and NP dimers in the PBS buffer outside the cell.

Figure R5 (Figure 7 B). The confocal microscopy images for the live (green) /dead (red) assays of HeLa cells after incubation with fixed NR dimers conjugated with Ce6 (without cell penetrating peptides labeled on the NP surface and denoted by **without TAT**) and irradiation under (A) LCP and (B) RCP for with 660 nm photons for 30 min (5 mW/cm^2). Scale bar $200 \mu\text{m}$.

Figure R6 (Figure 6 B). The confocal microscopy images for the live (green) /dead (red) assays of HeLa cells after incubation with fixed NR dimers conjugated with Ce6 (with cell penetrating peptides labeled on the NP surface and denoted by **with TAT**) and irradiation under (A) LCP and (B) RCP for with 660 nm photons for 30 min (5 mW/cm^2). Scale bar 200 μm .

Comment 3-2: Strongly recommends that you consider adding further discussion of the origins of this enhanced ROS effect to the manuscript.

Reply 3-2: Thank you for the constructive comments. The additional discussion of the ROS was added to the main text. Here we summarize the new points added to the manuscript regarding enhanced ROS production.

1. The evidence that rate of ROS generation is the key in understanding the biological effect of RCP and LCP photons for dimers with photodynamic therapy agents, can be substantiated by the observation of large difference of ROS under different illumination conditions and dispersion conditions replicating extra and intracellular conditions *ex vivo* (**Figure R8**). The experimental rate of ROS generations for NP and NR dimers in **Figure R8** changes synchronously with the killing rate in **Figures R1**. Note also that the chiral dimers not only as carry PpIX and Ce6 inside the cell but they also enhance ROS generation as can be seen from the curves in **Figure R8**. A large number of the hot electrons on NP and NR surface are produced that depending on circular polarization of the incident light, which could

further enhance the ROS generation.¹⁻³

2. In photodynamic therapy, the concentration of the photodynamic therapy agents and photo-generated ROS must be over the critical value to induce cell death.⁴ The dependence of killing rate on the transient concentration of ROS and therefore, on light absorption, is highly non-linear.⁴ The large difference between RCP and LCP photons is specific to the concentration of the photodynamic therapy agents and the chiral dimers. If one chooses the concentration of 4 μM for PpIX/ Ce6, almost all of the cell will be killed regardless of any light used. Similarly, if the low concentration (0.25 μM) of PpIX or Ce6 is used, the difference of death ratio between RCP, LCP, and other polarization will be small again (**Figures R7**).

In order to obtain the optimum therapeutic effect under CPL, the appropriate concentration of PpIX or Ce6 need to be used, that is 0.5 μM (**Figures R7**). Then, the difference in absorption of circular polarization of photons is amplified by the threshold phenomena.

Corresponding changes were made in the revised manuscript and highlighted in red.

Figure R7 (Figure 6 E, F). Viability of cells for different illumination conditions in the presence of various cellular loadings of PpIX (A) and Ce6 (B). The concentrations of photosensitizers were calculated in accordance with the average number of PpIX and Ce6 molecules attached to NP and NR conjugates.

Figure R8 (Figure 6 C,D) *Ex-vivo* singlet oxygen generation as a function of light exposure in model dispersions of NP (C) or NR (D) dimers conjugated to PpIX and Ce6 photosensitizers with variable and stationary (labeled as ‘fixed’) conformations (as in Figure 3), respectively. 5 mW/cm² light at 532 nm for 30 min was used for NP dimers; 5 mW/cm² light at 660 nm for 30 min was used with stationary NR dimers. ~94 molecules of PpIX were conjugated onto each NP and ~116 molecules of Ce6 were conjugated onto each NR. Intracellular conditions for C, D were experimentally reproduced by model cytosol medium as described in *Methods* section in the Supplementary Information, and error bars are given for standard deviation of 95%

REFERENCES:

- 1 Khaing Oo, M. K. *et al.* Gold nanoparticle-enhanced and size-dependent generation of reactive oxygen species from protoporphyrin IX. *ACS Nano* **6**, 1939-1947, (2012).
- 2 Hao, C. *et al.* Unusual Circularly Polarized Photocatalytic Activity in Nanogapped Gold–Silver Chiroplasmonic Nanostructures. *Adv. Funct. Mater.* **25**, 5816-5822, (2015).
- 3 Li, J. L., Day, D. & Gu, M. Ultra-Low Energy Threshold for Cancer Photothermal Therapy Using Transferrin-Conjugated Gold Nanorods. *Adv. Mater.* **20**, 3866-3871, (2008).
- 4 Cheng, L., Wang, C., Feng, L., Yang, K. & Liu, Z. Functional Nanomaterials for Phototherapies of Cancer. *Chem. Rev.* **114**, 10869-10939, (2014).

REVIEWERS' COMMENTS:

Reviewer #3 (Remarks to the Author):

The authors have sufficiently addressed the points raised in the previous round of review. At this point, the work appears to be suitable for publication.